# Die-off of plant pathogenic bacteria in tile drainage and anoxic water from a managed aquifer recharge site

Carina Eisfeld[1]*, Jan M. van der Wolf[2], Boris M. van Breukelen[1], Gertjan Medema[1,3], Jouke Velstra[4], Jack F. Schijven[5,6]

1 Faculty of Civil Engineering and Geosciences, Department of Water Management, Delft University of Technology, Delft, The Netherlands, 2 Wageningen University & Research, Wageningen, The Netherlands, 3 KWR Watercycle Research Institute, Nieuwegein, The Netherlands, 4 Acacia Water B.V., Gouda, The Netherlands, 5 Department of Earth Sciences, Environmental Hydrogeology Group, Utrecht University, Utrecht, The Netherlands, 6 Department of Statistics, Informatics and Modelling, National Institute of Public Health and the Environment, Bilthoven, The Netherlands

* carina.eisfeld@tudelft.nl

**Data Availability Statement:** All relevant data are within the manuscript and its Supporting information files.

## Abstract

Managed aquifer recharge (MAR) can provide irrigation water and overcome water scarcity in agriculture. Removal of potentially present plant pathogens during MAR is essential to prevent crop diseases. We studied the die-off of three plant pathogenic bacteria in water microcosms with natural or filtered tile drainage water (TDW) at 10 and 25˚C and with natural anoxic aquifer water (AW) at 10˚C from a MAR site. These bacteria were: *Ralstonia solanacearum* (bacterial wilt), and the soft rot Pectobacteriaceae (SRP) *Dickeya solani* and *Pectobacterium carotovorum* sp. *carotovorum* (soft rot, blackleg). They are present in surface waters and cause destructive crop diseases worldwide which have been linked to contaminated irrigation water. Nevertheless, little is known about the survival of the SRP in aqueous environments and no study has investigated the persistence of *R. solanacearum* under natural anoxic conditions. We found that all bacteria were undetectable in 0.1 mL samples within 19 days under oxic conditions in natural TDW at 10˚C, using viable cell counting, corresponding to 3-$\log_{10}$ reduction by die-off. The SRP were no longer detected within 6 days at 25˚C, whereas *R. solanacearum* was detectable for 25 days. Whereas in anoxic natural aquifer water at 10˚C, the bacterial concentrations declined slower and the detection limit was reached within 56 days. Finally, we modelled the inactivation curves with a modified Weibull model that can simulate different curve shapes such as shoulder phenomena in the beginning and long tails reflecting persistent bacterial populations. The non-linear model was shown to be a reliable tool to predict the die-off of the analysed plant pathogenic bacteria, suggesting its further application to other pathogenic microorganisms in the context of microbial risk assessment.

**Funding:** This research has been financially supported by the Netherlands Organisation for Scientific Research (NWO; Topsector Water Call 2016; project acronym AGRIMAR; contract number: ALWTW.2016.023; https://www.nwo.nl/onderzoeksprogrammas/topsector-water-call) ccwith co-funding from private partners Acacia Water B.V. (acaciawater.com), Broere Beregening B.V. (broereberegening.nl), and Delphy B.V. (delphy.nl). The funders had no role in study design, data collection and analysis, decision to publish, or preparation of the manuscript. The funders did not have any additional role in the study design, data collection and analysis, decision to publish, or preparation of the manuscript. The specific roles of the author with commercial affiliation is articulated in the 'author contributions' section.

**Competing interests:** The authors declare no competing interests in general and with any of the commercial affiliations. This commercial affiliation does not alter our adherence to PLOS ONE policies on sharing data and materials.

# 1 Introduction

Agricultural production requires intensification to satisfy the demands of the expanding world population. Nevertheless, crop production is often impaired by water shortages caused by droughts or the availability of fresh water [1]. For example, groundwater in coastal regions is not an option as irrigation source because it is often brackish [2]. Moreover, surface water can be unsuitable for irrigation as it may carry plant pathogens causing diseases such as brown rot to (seed) potatoes and ornamentals [3]. The implementation of water management strategies can reduce stress on water resources. Thereby, the potential of water reuse in agriculture gains attention with focus on the quality of the recycled water [4]. A nature based solution for water reuse is managed aquifer recharge (MAR) that collects excess tile drainage water after heavy rain events to store it in the subsurface. The stored water remains protected from evaporation and is available for irrigation or the management of the groundwater table in the field throughout the year. In this study, we investigate the die-off of bacterial plant pathogens in different water types from a MAR site to mimic the injection of contaminated water into the MAR system.

The *Ralstonia solanacearum* species complex (RSSC) comprises three different species, namely *R. solanacearum*, *R. pseudosolanacearum* and *R. syzigii*. They can cause bacterial wilt also known as brown rot, a highly destructive disease that affects more than 200 plant species [5, 6]. This pathogen complex has a quarantine status in the European Union [7]. In temperate climates, potato production is affected by *R. solanacearum* race 3 biovar 2 (phylotype II), a species adapted to cooler regions [8]. As a reaction to a high number of potato brown rot outbreaks in Europe in 1995, several measures were taken to control the disease, including seed testing. Although these regulations have shown some success, they still could not eradicate the disease completely [9]. In addition to latent infected seeds, other inoculum sources play a role in the sporadic infections, as the bacteria can survive in the environment, outside the potato plant. In the Netherlands, *R. solanacearum* has been detected in surface water over the past 20 years and brown rot disease outbreaks were directly linked to the use of contaminated surface water as irrigation source [10, 11]. As a result, an irrigation ban with surface water for seed potato production has been set in place which effectively reduced brown rot disease outbreaks [9]. Nevertheless, the measure reduced freshwater sources for irrigation.

Similar to bacterial wilt, blackleg and soft rot diseases caused by *Dickeya* and *Pectobacterium* species within the family of soft rot Pectobacteriaceae (SRP) have a negative impact on potato production [12]. The host range of the pectinolytic bacteria *Dickeya solani* and *Pectobacterium carotovorum* sp. *carotovorum* is not restricted to potato, they also affect other food crops and ornamental plants. For example, soft rot diseases pose a threat to the highly profitable flower bulb industry, whereas bacterial wilt caused by *R. solanacearum* has caused high damage in glasshouse ornamental crops [13]. Although SRP are well studied pathogens, measures to control the pathogens in the field are only partially effective. Latently infected seed material is the main cause for the dissemination of the pathogens. Still, soft rot and blackleg appear during the cropping season even if pathogen-free seed material grown via micropropagation was used [14]. The entry points are diverse as the bacteria have been observed in the environment, including plant debris, soil or waterways [15, 16]. Moreover, insects, nematodes, or aerosols emitted during field work contribute to the spread of SRP. In some studies, disease outbreaks were related to the usage of contaminated irrigation water [17]. Kastelein et al. [18] reported that spray inoculation of potato leaves with $10^2$ CFU/mL *D. solani* resulted in diseased plants in greenhouse experiments.

A solution to provide irrigation water free of pathogens can be MAR, where excess freshwater is stored in the subsurface and serves as an irrigation reservoir [19]. In this research, we

investigate a MAR system which uses aquifer storage, transfer and recovery (ASTR) technology that injects collected fresh tile drainage water (TDW) into an anoxic brackish aquifer. The TDW is excess rain water, that percolates through the upper soil layer into drains, buried below the agricultural field. Depending on the agricultural practices and period of the year, the chemical and biological composition of the TDW may change. For example, during the cropping season fertilizers are applied which can lead to increased nitrate concentrations in the TDW. In ASTR the injected water travels through the soil towards the extraction well. We expect that this natural soil filtration step removes agropollutants, including bacterial pathogens, in different removal processes. The major processes are die-off in the water phase, irreversible attachment to sediment particles, straining, as well as die-off through predation and competition with other microorganisms [20]. Very little information is available about the removal efficiency of pathogens during MAR as the potential of subsurface treatment has been poorly recognized [21]. The irrigation water can be considered safe, if the pathogen load of the recycled water is below the threshold inoculum level which does not lead to the infection of crop plants. As part of the risk estimation of reusing TDW for agricultural irrigation, we need to determine the removal of the pathogens from the system.

Few studies investigated the die-off of *R. solanacearum* in natural water types such as river water [22], agricultural drainage water [23], or drainage water leaching from rock wool in a rose-growing greenhouse [24]. The results of these studies showed that temperature and the natural microbiota have a significant effect on the die-off of *R. solanacearum* in non-sterile microcosms compared with sterile conditions. However, no study was found that describes the die-off of the bacterium under anoxic conditions in natural water, which is highly relevant for their fate in saturated soil conditions, as found in the aquifer or deeper soil layers where oxygen is limited. To our knowledge, only one publication examined the inactivation of *D. solani* and *P. carotovorum* in sterile waters, but not in waters under natural conditions [25].

The aim of this study was to investigate the die-off of *R. solanacearum*, *D. solani*, and *P. carotovorum* sp. *carotovorum* in natural TDW and anoxic aquifer water from an agricultural field and to develop a die-off model. Knowledge about the survival of these plant pathogens in the agro-ecosystem is relevant for a successful management of the disease. The model can be used in microbial risk assessment to predict the die-off of the pathogens in natural waters to minimize the risks of irrigation related disease outbreaks. For a safe MAR application, the input for the risk estimation requires the precise description of the bacteria's die-off. So far, the die-off of *R. solanacearum* has been described using log-linear kinetics. Linear models may describe the die-off of the bacteria insufficiently by over- or underestimating their persistence, especially over prolonged time periods. A bacterial population may vary in resistance to environmental stresses and, therefore, die-off follows a non-linear pattern instead of a log-linear (first-order rate) [26]. It has been shown for *R. solanacearum* that different subpopulations are present during the die-off in microcosms. Moreover, the bacteria underwent morphological changes or entered into the viable but non-culturable (VBNC) state [23, 27]. For a better prediction of the bacterial die-off, we applied a non-linear Weibull + tail, a Weibull, and a log-linear model, modified after Albert and Mafart [28], and selected the best model. Finally, the selected die-off models are a crucial component in risk assessments that aim to develop guidelines for the safe application of recycling water systems such as MAR.

## 2 Material and methods

### 2.1 Bacterial strains and growth conditions

*R. solanacearum* race 3 biovar 2 (phylotype II) strain IPO-1828, *D. solani* IPO-2266 and *P. carotovorum* sp. *carotovorum* IPO-1990 were used in this study. They were received from the

working collection at Wageningen UR. Strains were kept at -80˚C using the multi-purpose protect cryobeads system (Technical Services Ltd). The strain of *R. solanacearum* used in this study is naturally resistant to rifampicin and was re-isolated from tomato plants infected with the Dutch strain IPO-1609, originally isolated from potato. It was grown on non-selective Yeast Peptone Glucose Agar (YPGA) [29], prepared with 5 g/L yeast extract, 5 g/L peptone, 10 g/L glucose, 15 g/L agar, and supplemented with rifampicin (50 mg/L). Liquid cultures were prepared in Casamino acid Peptone Glucose (CPG) broth [30] composed of 1 g/L casamino acids, 10 g/L peptone, and 5 g/L glucose. *D. solani* and *P. carotovorum* sp. *carotovorum* were both naturally resistant to streptomycin and were grown on non-selective Tryptone Soya Agar (TSA; Oxoid; Thermo Fisher Scientific) supplemented with streptomycin (100 mg/L). Liquid cultures of these bacteria were prepared in LB (Luria Bertani) medium. Duchefa Biochemie (Haarlem, The Netherlands), Sigma-Aldrich (St. Louis, MO), and Fisher Scientific (Hanover Park, IL) were our chemical suppliers.

## 2.2 Water samples

The research site as part of the Spaarwater project [31] initiated by Acacia Water B.V. is located on a 1.5 ha field in Borgsweer, situated in the province of Groningen, the Netherlands, near the Wadden sea (53.2945693 N, 7.0045595 E). Access to the field site was granted by Acacia Water B.V. and the farmer. The marine clay topsoil is laying on a thick aquitard layer of 8 m depth consiting of clay and peat, and is optimal for the cultivation of (seed) potatoes. The irrigation with surface water is forbidden by Dutch law with the risk of spreading the brown rot causing *R. solanacearum*. The ASTR system, as shown in Fig 1, is used to store fresh TDW beneath the aquitard layer, in the underlying, confined unconsolidated sandy aquifer stretching over a depth of -8 m to -30 m below surface level. The aquifer conists of layers originating from different geological formations, resulting in different hydraulic conductivities (4–45 m/day). After infiltration, the fresh water is available for later abstraction and can be used as irrigation water in dry summer periods. Below the parcel, a network of drainage pipes is buried which are connected to a collection drain that discharges into a built concrete reservoir (volume is ca. 1 m$^3$). In this reservoir, the electrical conductivity (EC) of the TDW is continuously sensed as measure of salinity. The water is used for infiltration into the aquifer if the EC does not exceed a set value (EC = 1700 μS/cm) needed for the irrigation of the selected crops. Otherwise, the TDW is discharged into the surface water system along the agricultural fields. As the pilot was no longer in operation, the last infiltration event dated back to more than a year ago. The tile drainage system was still operable and collected TDW which was automatically discharged into the surface water canal neighboring the field. The microbial aquifer community in the aquifer was therefore not influenced by any recent TDW injection. Two water samples were collected at this research site for the water microcosm experiments (described later).

**Sampling.** All water samples were collected in autoclaved 1- or 2-L glass bottles and completely filled before closing. Water quality parameters such as pH, EC, oxygen concentration and water temperature were measured in the field using a portable multimeter with respective probes (Odeon, Aqualabo, France). In the laboratory, Hach Lange kits were used to measure nitrate, phosphate, chemical oxygen demand (COD), and ammonium. The kits are based on colorimetric assays and the concentrations were assessed using a VIS-spectrophotometer (DR3900, Hach Lange, Germany). To be able to collect fresh TDW, we awaited a rain event which would allow the flushing of the drainage system. On a first field visit in April 2018, oxic TDW water was collected from the TDW collection point (concrete reservoir) of the MAR system. During a second visit in December 2018, anoxic water from the aquifer (9–10 m below surface level) was pumped from a monitoring well with a 1 m long filter screen,

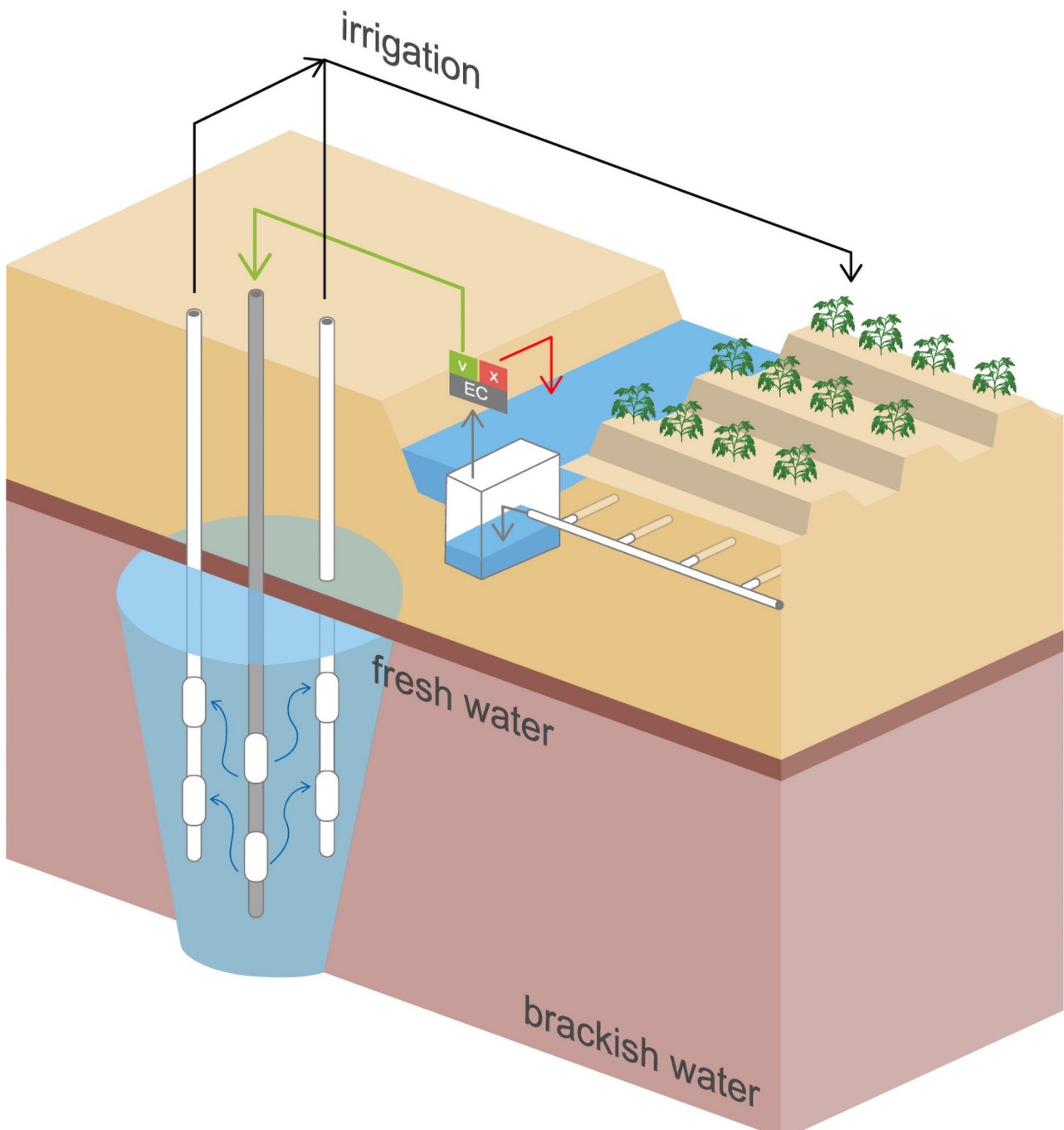

**Fig 1. Scheme of a managed aquifer site.** Schematic representation of an agricultural field connected to a managed aquifer recharge site. The site is designed as an aquifer, storage, transfer, and recovery (ASTR) system. Excess rain water reaches the tile drainage system buried at 70 cm depth. The collection drain terminates into a concrete reservoir where the EC of the tile drainage water is measured. If the EC is below a set threshold value, the water is infiltrated via the injection well (depicted in gray). From there, the water travels through the sandy aquifer to the extraction wells (depicted in white) and can be used for irrigation.

using a peristaltic pump (Watson Marlow 704U/R) at a flow rate of 12 L/min. The tubing system was cleaned before sampling using a 10 times diluted solution of commercial bleach, followed by pumping for 10 min from the selected well to replace the standing water in the well at least 3 times. The anoxic water was filled into 2 L Duran glass bottles that were priorly flushed with $N_2$ gas. The tube connected to the pump was placed at the bottom of the bottle. The bottle was then filled from bottom to top, while keeping the tube inside the water to limit the introduction of air. After filling the bottles completely with water, they were first closed with a black rubber stopper, and then closed with a plastic screw cap with aperture, which allows to take a water sample with a needle and syringe while maintaining anoxic conditions.

## 2.3 Die-off experiments with water from a MAR site

The die-off experiments were conducted with naturally oxic tile drainage water and with anoxic aquifer water.

As summarized in Table 1, the waters have been treated in four different ways before the bacterial pathogens were added: (i) natural oxic TDW (containing nitrate) or anoxic aquifer water (almost without nitrate) without treatment, maintaining the whole microbiota; (ii) 0.22 μm filtered TDW, to remove most of the microbiota; (iii) autoclaved TDW to inactivate all microbiota as control; (iv) natural aquifer water with the addition of 50 mg/L $NO_3^-$ (added as sodium nitrate); to assess the possible influence of nitrate, present in TDW as a result of agricultural practices. This nitrate concentration was chosen as it is representative for concentrations found in Dutch surface waters receiving TDW. Nitrate will be present in the aquifer until the infiltrated TDW is denitrified. For both the natural microcosms of oxic TDW and anoxic AW a non-inoculated control microcosm was prepared. The oxic TDW microcosms were incubated at 10˚C or 25˚C in the dark while shaking (150 rpm). Anoxic experiments were conducted only with natural aquifer water at one temperature (10˚C). Experiments with natural aquifer water where nitrate was added, were only tested with *R. solanacearum* and *D. solani*. The temperatures were chosen as they represent the nearly constant temperature in the aquifer (10˚C); while 25˚C reflects infiltration of rainfall events during the warmer summer months. Moreover, the latter covers a temperature that is more representative of tropical regions, where these bacteria also play an important role as disease causing organisms.

For experiments under natural conditions, the water samples were used 1–4 days after collection to avoid changes in the microbial composition. For die-off experiments in filtered water, TDW was filtered using a sterile 0.22 μm pore-size cellulose acetate filter membrane (Sartorius, Germany). Autoclaved conditions were achieved by pressure-sterilization at 121˚C for 20 min. For die-off experiments in 0.22 μm filtered or autoclaved conditions, the samples were stored at 4˚C until usage. Note, filtered and autoclaved experiments are artificial conditions not representative of natural conditions during MAR. They served as control experiments to analyze the effect of microbiota on the pathogens as well as the general persistence of pathogens in sterile water.

The inoculation suspensions were prepared by growing *R. solanacearum* for 15 h in 5 mL CPG broth and *D. solani* or *P. carotovorum* sp. *carotovorum* for 12 h in 5 mL LB liquid medium. Culture conditions were 25˚C and 150 rpm, after which the grown cultures were harvested by centrifugation (3500 x *g*, 20 min at room temperature) followed by washing and resuspending the pellet in 5 mL a quarter strength Ringer's solution (Sigma-Aldrich; St. Louis, USA). This pelleting and washing step was repeated twice to remove any excess broth. The bacterial suspension was then diluted to reach an optical density of 0.1 at 600 nm representing a concentration of $10^8$ CFU/mL which was also confirmed by dilution-plating. For the aerobic experiments, 1 L of each microcosm solution was prepared in a sterile glass flask and 0.1 mL of

**Table 1. Experimental conditions for the bacterial inactivation study in different water types.**

| Treatment | natural | 0.22 μm filtered | autoclaved | natural | natural + 50 mg/L NO$_3$ |
|---|---|---|---|---|---|
| Water type | | tile drainage | | | aquifer |
| Redox | | Oxic | | | Anoxic |
| Temperature [˚C] | | 10 or 25 | | | 10 |
| Nitrate* | | 41 mg/L | | | < 0.5 mg/L |
| Inoculation concentration of bacterial pathogen | | | $10^4$ CFU/mL | | |

* average concentrations at Borgsweer pilot site, 2016–2017 [31]

$10^8$ CFU/mL bacterial suspension was added to reach a final concentration of $10^4$ CFU/mL. From this suspension, 200 mL were transferred to 250 mL Duran glass bottles. This procedure could not be used in the anoxic experiment as the introduction of oxygen had to be excluded. There, 180 mL serum bottles (Wheaton Scientific, Millville, USA) were flushed with N$_2$ gas, closed with a butyl rubber stopper and a metal crimp, and autoclaved. With a 50 mL syringe attached to a needle, a total of 100 mL of the aquifer water was transferred to the experimental flask. Before the transfer, the syringe was flushed three times with N$_2$ gas to avoid oxygen contamination during the transfer. Each flask had to be inoculated individually. The inoculation solution was prepared as described previously; but here, 0.1 mL of $10^7$ CFU/mL bacterial suspension was added to the 100 mL microcosm to reach a final concentration of $10^4$ CFU/mL. The concentration of $10^4$ CFU/mL as inoculum was chosen as the pathogen concentrations in natural water systems are often found to be very low. Wenneker et al. [10] detected *R. solanacearum* in ditches next to agricultural fields at concentrations ranging between $10^2$–$10^5$ CFU/mL. Higher inoculum density may give the introduced pathogenic bacteria a competitive advantage against other microorganisms. Furthermore, mimicking more realistic concentrations with lower starting concentrations in the laboratory is not convenient as high volumes of water are necessary for the detection. The chosen starting concentration is a compromise and allows to follow the inactivation over a sufficiently long time period and the collection of sufficient data points for establishing the type of die-off kinetics by modelling. All microcosms were prepared in duplicates and the bacterial numbers were evaluated by dilution plating in duplicates per time point.

## 2.4 Enumeration of bacteria from water microcosms

Sampling from the microcosms was performed regularly after inoculation until the last sampling point was reached, which was at the detection limit of 3–10 CFU/mL or below the detection limit when no colonies were culturable anymore. *R. solanacearum* was enumerated on semi-selective medium South Africa (SMSA) [32] supplemented with 50 mg/L rifampicin to suppress the growth of background bacteria. *D. solani* and *P. carotovorum* sp. *carotovorum* were incubated on the selective double layer-crystal violet pectate (DL-CVP) medium [33] supplemented with 100 mg/L streptomycin, where the bacteria form distinct cavities in the upper layer of the medium. The selective media have been chosen as they showed a higher recovery rate from the environmental water samples in comparison to TSA supplemented with the respective antibiotic (data not shown). Moreover, they better suppress the growth of non-specific bacteria. The initial concentration of indigenous culturable bacteria in the natural oxic and anoxic water samples was assessed by plating 0.1 mL on non-selective TSA and low-nutrient Reasoner's 2A (R2A) agar (Oxoid; Thermo Fisher Scientific) and incubated at 25˚C. Anaerobic water samples were incubated anaerobically 25˚C using Oxoid AnaeroGen sachets

(Oxoid; Thermo Fisher Scientific) to create an anaerobic atmosphere in a closed container. Additionally, total numbers of culturable bacteria were enumerated in all oxic TDW microcosms from both temperatures at the end of the die-off period of the SRP. For sampling, microcosms were mixed by vigorous shaking, then 0.1–0.3 mL of water sample was taken for detection of the bacteria by dilution plating on the respective medium in duplicates. Plates were incubated three to four days at 28°C before counting. Water samples from anoxic microcosms were sampled through the rubber septum with a syringe and needle to maintain anaerobic conditions in the microcosm; afterwards the plates were incubated aerobically. We define die-off as the process that renders bacterial cells not culturable anymore.

## 2.5 The die-off model

As a first step, the pathogen concentration defined as culturable cells per mL was plotted against time. The resulting graph gave insights into the course of the die-off. Three types of die-off models were considered, as shown in Fig 2.

Our results showed that in most cases, the bacteria did not follow a linear die-off behavior. Moreover, the initial bacterial population stayed stable over a period of time resulting in a curve with a shoulder (see Fig 2, model W+t and W). Another feature of the curve can be a tail-shape which represents a persistent population at the end of the experiment (see Fig 2, model W+t). To account for these curve characteristics, a model based on the work of Albert and Mafart [28] to describe microbial die-off was chosen. In the further text we refer to it as the Weibull + tail (W+t) model:

$$C_t = (C_0 - C_{res}) \, e^{-(\alpha t)^{\beta}} + C_{res} \qquad\qquad (1 : W + t)$$

where $C_t$ is the bacterial concentration in CFU/mL at time $t$, $C_0$ the initial bacterial concentration in CFU/mL (at time $t = 0$), $C_{res}$ [CFU/mL] is the residual bacterial population at the end of the observation period, and $t, \alpha, \beta > 0$. $\alpha$ [1/day] is a scale parameter and $\beta$ [–] a shaping parameter to display convexity of a curve if $0 < \beta < 1$, or simulates a shoulder effect when $\beta > 1$. $C_0$, $C_{res}$, $\alpha$ and $\beta$ are unknown parameters that have to be estimated.

The Weibull model [34] describes microbial inactivation as a statistical distribution of die-off times. Thereby, it accounts for the heterogeneity of a bacterial population where different subpopulations exist with different resistances to the exposed stress. Consequently, $\alpha$ is not a single die-off rate constant that can be applied to the whole bacterial population, but the mean of the distribution describing the death times. Van Boekel [26] concludes that the Weibull is an empirical model that can still be linked to physiological effects, where $\alpha$ relates to microbial inactivation, while the dependency of $\beta$ can be linked to abiotic and biotic conditions and the adaptation of the bacteria to the external stress. Furthermore, the authors state that if $\beta < 1$, the remaining cells are better adapted to stress and have less probability of dying. In contrast, $\beta > 1$ implies that the cells are increasingly damaged and susceptible to stress. The Weibull + tail model can be reduced to the non-linear Weibull (W) model if there is no tailing and the parameter $C_{res} = 0$:

$$C_t = C_0 \, e^{-(\alpha t)^{\beta}} \qquad\qquad (2 : W)$$

A further reduction of the Weibull model results in a log-linear model (L) if $\beta = 1$, corresponding to first order die-off. As consequence, the die-off probabilities are not time dependent anymore, and $\alpha$ is a constant rate parameter. Lastly, this implies that the bacterial population is homogenous and the cells are equally susceptible to environmental stresses.

$$C_t = C_0 \, e^{-(\alpha t)} \qquad\qquad (3 : L)$$

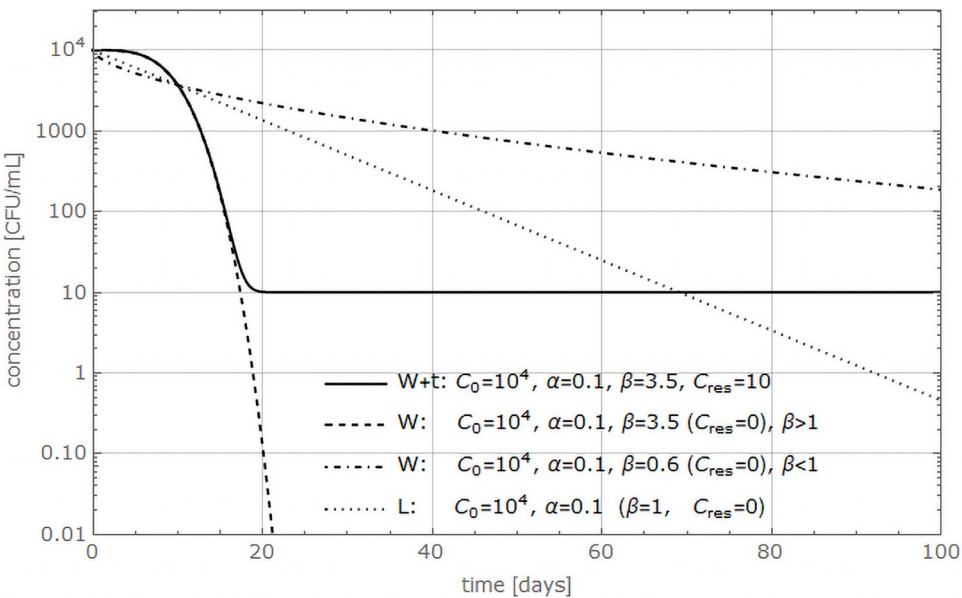

**Fig 2. Die-off models.** Different die-off models were simulated with given parameter estimates (see values in the graph). The pathogen concentration in CFU/mL is plotted against time in days. W+t stands for Weibull + tail, W for Weibull, and L for log-linear model.

A visual selection of the model just by looking at the curve shape is useful, but fails to exactly determine the best model fit. All three models were applied to fit the experimental data and the best model was chosen using the dimensionless Akaike Information Criterion (AIC) [35]. The AIC considers the goodness of fit and parsimony of the models. For our experiments, the model with the smallest AIC value was therefore chosen as best model.

**Statistical analysis.** The data was analyzed using the *R* software [36] (version (v.) 1.2.1335) with different statistical packages: (i) 'dplyr' to organize the data (*v. 0.8.3*, [37]); (ii) 'nlstools' for the statistical analysis and non-linear modelling (*v. 1.0–2*, [38]); (iii) 'investr' to plot the non-linear model including prediction intervals (*v. 1.4.0*, [39]) and (iv) 'nlshelper' to use F-tests or ANOVA as analysis if the estimated coefficients were different between groups (*v. 0.2*, [40]). Non-linear modelling using *R* is well described by Duursma [41], as well as the utilization of the 'nlstools' package by Baty et al. [42]. The package also allows to examine the model fit by analyzing the distribution of the residuals. The R code used in this study is described in the S1 File. The data frame consisted of a total of 986 observations with 4 to 22 data points per experiment. The various treatments of the microcosms, as well as the temperature and redox conditions were evaluated as categorical variables or factors.

## 3 Results

### 3.1 Description of field site/initial conditions

In natural TDW at 10 and 25°C, the indigenous culturable bacterial levels in the control microcosms were $10^2$ CFU/mL grown on TSA and R2A, and reached 9.6–9.8 x$10^3$ CFU/mL after 16 days, grown on TSA or R2A respectively. In the control microcosms of anoxic AW with or without nitrate, the concentration of culturable indigenous bacteria was about 2.3 x$10^2$–4 x$10^3$ CFU/mL, grown on TSA or R2A respectively. S1 Table shows an overview of the enumeration of culturable microorganism on TSA and R2A present in the native TDW and

AW. No indigenous bacteria were culturable on SMSA supplemented with rifampicin. Only a few unspecific bacteria were growing on the upper layer of DL-CVP supplemented with streptomycin that were clearly distinguishable from the cavity forming soft rot bacteria. Initially, no bacteria were detected when plating the filtrated TDW on TSA. However, at the end of the experiment with SRP at 25°C, some colonies appeared ($10$–$10^2$ CFU/mL) when plating the filtered TDW, as maybe a few bacteria passed the filter and recovered during the experiment. Long-term persistence of the pathogens was demonstrated in the autoclaved microcosms where optimal conditions (available nutrients, no competition with other microorganism) resulted in growth of the pathogens in the beginning of the experiments. In the autoclaved microcosms the physicochemical composition of the water is changed during the heating process. In contrast, the filtration only removes the biotic fraction from the water but leaves the abiotic conditions unchanged. This allows to study the influence of the biotic factors in oxic microcosms when comparing natural and 0.22 μm filtered TDW. Table 2 gives an overview of the water quality at the MAR pilot site. The measured values of the chemical oxygen demand (COD) represent the total organic compounds in the water and they are representative for the range found in unpolluted surface waters (around 20 mg/L). Overall, the TDW is oxic and contains nitrate levels in the order of 50 mg $NO_3$/L. The AW is anoxic and almost free from nitrate (see also [31]). The temperature of the TDW varies with the season, while the AW remains at a constant temperature of around 10°C. Total phosphate and nitrate concentrations did not change after 0.22 μm filtration or autoclaving of the TDW.

## 3.2 Bacterial die-off in water microcosms

The die-off of three bacterial plant pathogens was studied in microcosms filled with natural or treated water at different temperatures. A total of 17 datasets were evaluated and the fitted die-off curves are shown in Fig 3. The experiments were concluded when the limit of detection of 3–10 CFU/mL was reached (visualized as horizontal dotted line in Fig 3) or the bacteria were no more detectable by plating (below the detection limit, as marked with an asterisk in Fig 3). To select the best model fit, the Akaike information criterion (AIC) values (S2 Table) of the three models were compared after fitting the data, and the model with the smallest AIC is chosen. The Weibull + tail model, as well as the Weibull model were both selected eight times as the best model. Only in one case, the die-off of *D. solani* in filtered TDW at 25°C, the log-linear model was selected as the best model, although the AIC values for the Weibull and log-linear model were very similar (see S2 Table). Nevertheless, the Weibull + tail model could only be fitted to the data if the bacterial concentration remained constant at the end of the die-off period, resulting in a tail shape in the plotted curve. The results of experiments in autoclaved microcosms are shown in S1 Fig. There, the bacteria started to grow at 10 and 25°C reaching a concentration of $10^6$ CFU/mL followed by a slow decline until they reached a concentration of at least $10^3$ CFU/mL which lasted until the end of the observation period (150 days). The only exception from this trend was the die-off of *R. solanacearum* in the autoclaved TDW at 10°C which rendered non-culturable after a period of approximately 30 days. In the further subsections, the die-off of each bacterium is described in detail and Table 3 shows an overview of the estimated model parameter values. The ANOVA analysis revealed that our tested treatments resulted in significantly different bacterial die-offs (P < 0.05). In anoxic AW, the addition of nitrate did not have a significant effect (P = 0.23) on the die-off of *R. solanacearum*. In contrast, the addition of nitrate had a significant effect on the die-off of *D. solani* under the same conditions (P = 0.003). There, the bacterial die-off was slower (56 days) if no nitrate was added, as nitrate addition accelerated the die-off period (48 days).

**Table 2. Water quality of natural waters at a managed aquifer recharge site.**

| Water quality parameters | | Tile drainage water | | Aquifer water |
| --- | --- | --- | --- | --- |
| | | **2018/04/17** | **2018/12/10** | **2018/12/10** |
| measured in the field | Electrical Conductivity [μS/cm] | 1940 | 1490 | 1150 |
| | pH | 6.7 | 6.9 | 7.3 |
| | Temperature [˚C] | 9.5 | 7.4 | 8.3 |
| | $O_2$ [mg/L] | 3.35 | 8.42 | 0.33 |
| measured in the laboratory | Chemical Oxygen Demand (COD) [mg $O_2$/L] | 21.2 | 29.5 | 20.6 |
| | Nitrate ($NO_3$) [mg/L] | 24.6 | 70.3 | 0.4 |
| | Total phosphate ($PO_4$) [mg/L] | 0 | 0.02 | 0.28 |
| | Ammonium ($NH_4$) -N [mg/L] | 0 | 0.17 | 0.94 |

### 3.3 Ralstonia solanacearum

The population of *R. solanacearum* in natural TDW declined by 3-$\log_{10}$ within 19 days at 10˚C, and within 25 days at 25˚C (Fig 3, R1 & R2). The die-off in 0.22 μm filtered TDW took longer in comparison to the untreated conditions because most of the indigenous microbiota was removed during the filtration. Therefore, the faster die-off in natural water microcosms can be attributed to the presence of the microbiota because the abiotic factors stayed the same. Furthermore, the die-off in filtered TDW underlined the temperature sensitivity of *R. solanacearum*. While the detection limit in filtered TDW at 10˚C was reached after 25 days, the die-off in the same water type at 25˚C was more than two times longer. A similar observation was made in autoclaved TDW at 10˚C, where *R. solanacearum* is no longer culturable after a period of 30 days, even though the bacteria were culturable for a long period of 150 days at 25˚C (see S1 Fig). In filtered TDW at 25˚C, the bacteria persisted at a low concentration of ca. 10 CFU/mL for up to 60 days creating a long tail-shape in the plotted curve (see Fig 3, R4), which can also be observed under anaerobic conditions for a period of ca. 25 days (Fig 3, R5 & R6). Surprisingly, the die-off of *R. solanacearum* under anoxic conditions took about 45 days and was two times slower than in natural TDW despite the low temperature of 10˚C and absence of oxygen. To summarize, regardless of the water microcosm or treatment, and a starting concentration of $10^4$ CFU/mL, the concentrations of *R. solanacearum* decreased by 3-$\log_{10}$ within the first 25 days of the experiments but resulted in many cases in a persisting population (Fig 3, R2, R4-R6). In comparison to *D. solani* and *P. carotovorum* sp. *carotovorum*, the plotted curves of *R. solanacearum* have the most prominent shoulder and tailing phenomena. Four out of the six datasets are fitted best with the Weibull + tail model.

### 3.4 Soft rot Pectobacteriaceae

The inactivation behavior of the soft rot Pectobacteriaceae (SRP) *P. carotovorum* sp. *carotovorum* and *D. solani* were very similar as seen in Fig 3, D & P(1–6). We observed the quickest die-off of the tested SRP in natural oxic TDW at 25˚C, where the bacteria were culturable for four days (Fig 3, D2 & P2); and for a maximum of 12 days at 10˚C (Fig 3, D1 & P1). In contrast, the die-off in filtered TDW was decelerated at least three times compared with untreated TDW. In filtered TDW at 10˚C, the bacteria showed a comparably slow die-off: the populations were stable (shoulder phenomenon, Fig 3, D3 & P3) for about 30 days, after which the concentrations decreased continuously, reaching the detection limit after 50 days. The die-off in the same water type at 25˚C was much faster; within 17 days (Fig 3, D4 & P4). The faster die-off at 25˚C compared to 10˚C was noticeable in untreated and in filtered TDW, implying lower resistance of the SRP at a higher temperature. Still, the die-off in autoclaved TDW was

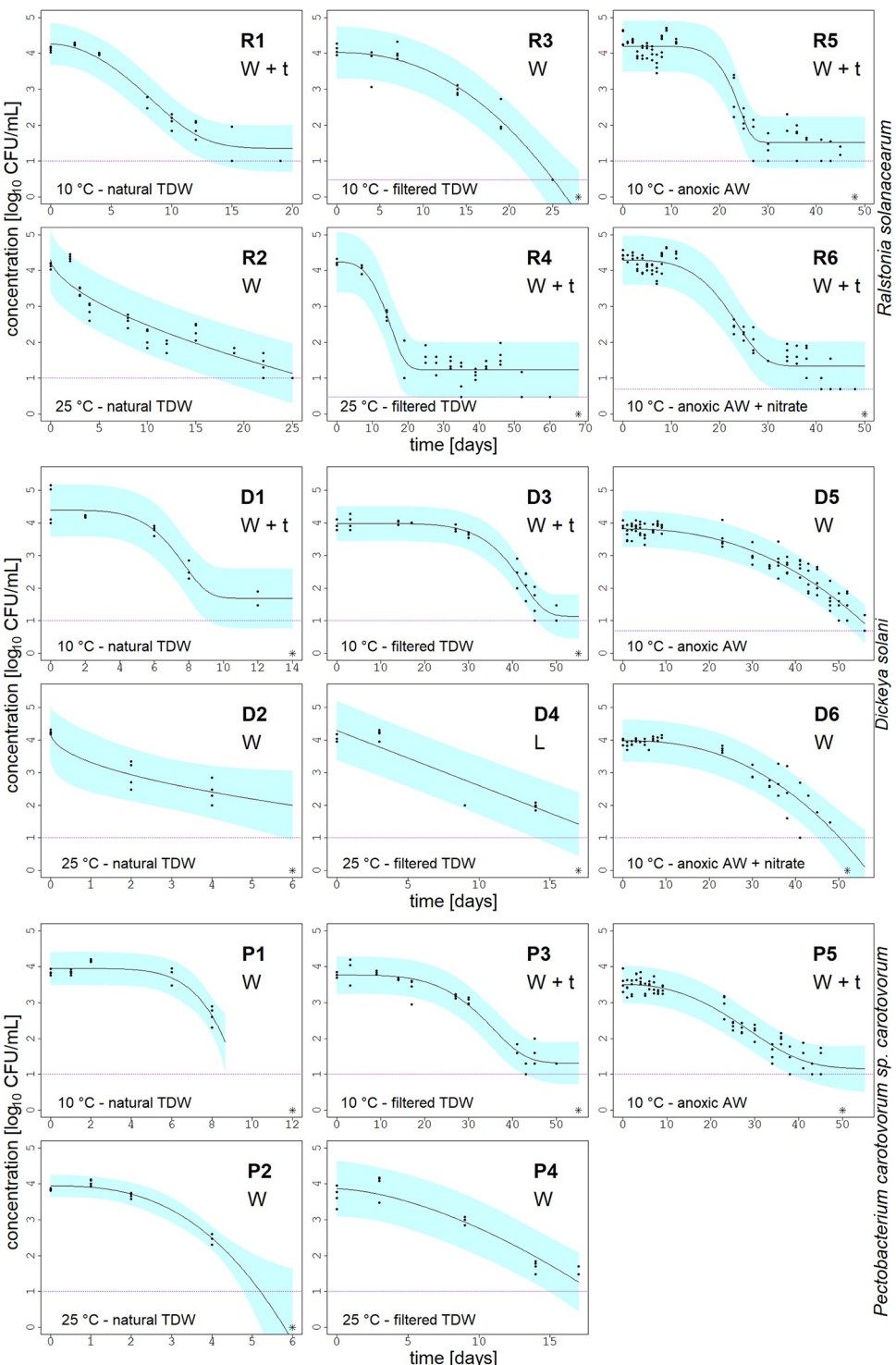

**Fig 3. Die-off curves of plant pathogens.** Die-off of *Ralstonia solanacearum*, *Dickeya solani*, and *Pectobacterium carotovorum* sp. *carotovorum* in microcosms under varying conditions, shown as $\log_{10}$ [CFU/mL] vs. time [days]. The experimental conditions are displayed in the lower left corner of each graph. Points represent the plate counts in duplicate of two microcosms per treatment. The solid black line displays the model fitted to the data. The blue area depicts the 95% prediction interval of the model and the purple dotted line is the detection limit. The model applied used to fit the data is shown in the right corner of each graph: W+t stands for Weibull + tail, W for Weibull and L for log-linear model. The asterisk sign accounts for the last measurement point, where no more viable colonies were detectable.

**Table 3. Model parameter estimates and their 95% Gaussian confidence intervals shown in brackets.**

| dataset | | [°C] | treatment | redox | Log$_{10}(C_0)$ [CFU/mL] | $C_0$ [CFU/mL] | Log$_{10}(C_{res})$ [CFU/mL] | $C_{res}$ [CFU/mL] | $\alpha$ [1/day] | $\beta$ [−] |
|---|---|---|---|---|---|---|---|---|---|---|
| *Ralstonia solanacearum* | **Weibull + tail** | | | | | | | | | |
| | R1 | 10 | natural | oxic | 4.3 [4.0; 4.5] | 1.8 x10$^4$ | 1.4 [1.1; 1.7] | 22 | 0.25 [0.14; 0.36] | 1.7 [1.0; 2.4] |
| | R4 | 25 | filtered | oxic | 4.3 [3.9; 4.6] | 1.7 x10$^4$ | 1.2 [1.1; 1.4] | 17 | 0.12 [0.06; 0.16] | 2.5 [1.0; 4.0] |
| | R5 | 10 | natural | anoxic | 4.2 [4.1; 4.3] | 1.6 x10$^4$ | 1.5 [1.4; 1.7] | 33 | 0.05 [0.04; 0.06] | 5.6 [2.2; 9.1] |
| | R6 | 10 | natural + NO$_3$ | anoxic | 4.3 [4.2; 4.4] | 1.9 x10$^4$ | 1.3 [1.2; 1.5] | 21 | 0.07 [0.04; 0.08] | 3.1 [1.6; 4.5] |
| | **Weibull** | | | | | | | | | |
| | R2 | 25 | natural | oxic | 4.3 [3.9; 4.7] | 2.0 x10$^4$ | | | 1.10 [-0.09; 2.28] | 0.6 [0.4; 0.8] |
| | R3 | 10 | filtered | oxic | 4.0 [3.8; 4.3] | 1.1 x10$^4$ | | | 0.10 [0.07; 0.13] | 2.2 [1.5; 3.0] |
| *Dickeya solani* | **Weibull + tail** | | | | | | | | | |
| | D1 | 10 | natural | oxic | 4.4 [4.1; 4.7] | 2.4 x10$^4$ | 1.7 [1.2; 2.2] | 49 | 0.18 [0.13; 0.22] | 3.9 [1.4; 6.4] |
| | D3 | 10 | filtered | oxic | 4.0 [3.9; 4.1] | 9.6 x10$^3$ | 1.2 [0.8; 1.6] | 14 | 0.03 [0.02; 0.03] | 5.5 [3.5; 7.4] |
| | **Weibull** | | | | | | | | | |
| | D2 | 25 | natural | oxic | 4.2 [3.9; 4.6] | 1.7 x10$^4$ | | | 4.61 [-8.63; 17.85] | 0.5 [0; 1.0] |
| | D5 | 10 | natural | anoxic | 3.8 [3.7; 3.9] | 6.6 x10$^3$ | | | 0.04 [0.03; 0.04] | 2.2 [1.8; 2.7] |
| | D6 | 10 | natural + NO$_3$ | anoxic | 4.0 [3.9;4.1] | 9.6 x10$^3$ | | | 0.04 [0.03; 0.05] | 2.4 [1.7; 3.1] |
| | **First-order** | | | | | | | | | |
| | D4 | 25 | filtered | oxic | 4.3 [4.0; 4.6] | 2.0 x10$^4$ | | | 0.39 [0.29; 0.47] | |
| *Pectobacterium carotovorum* sp. *carotovorum* | **Weibull + tail** | | | | | | | | | |
| | P3 | 10 | filtered | oxic | 3.8 [3.7; 3.9] | 5.8 x10$^3$ | 1.3 [1.1; 1.6] | 20 | 0.04 [0.03; 0.04] | 3.3 [2.3; 4.3] |
| | P5 | 10 | natural | anoxic | 3.5 [3.4; 3.6] | 3.2 x10$^3$ | 1.2 [0.9; 1.5] | 14 | 0.06 [0.04; 0.07] | 1.9 [1.3; 4.3] |
| | **Weibull** | | | | | | | | | |
| | P1 | 10 | natural | oxic | 4.0 [3.8; 4.1] | 8.9 x10$^3$ | | | 0.15 [0.13; 0.17] | 5.6 [2.4; 8.9] |
| | P2 | 25 | natural | oxic | 3.9 [3.8; 4.1] | 8.7 x10$^3$ | | | 0.39 [0.3; 0.47] | 2.7 [1.5; 3.8] |
| | P4 | 25 | filtered | oxic | 3.9 [3.6; 4.2] | 7.3 x10$^3$ | | | 0.18 [0.08; 0.26] | 1.6 [0.9; 2.4] |

similar at both temperatures. Interestingly, the die-off of *D. solani* in filtered TDW at 25°C is the only dataset that follows first-order inactivation. The datasets of *D. solani* in anoxic AW at 10°C could both be fitted with the Weibull model (Fig 3, D5 & D6), while the die-off of *P. carotovorum* sp. *carotovorum* in anoxic AW was fitted with the Weibull + tail. The tailing phenomenon that was clearly visible in the inactivation curves of *R. solanacearum* (e.g. Fig 3, R4), is shorter in the curves of the SRP (Fig 3, D1 & D3, P3 & P5). As with *R. solanacearum*, the die-off of the SRP can be better explained by using the non-linear regression models.

### 3.5 Parameter estimation

The parameter estimates of the best fitting model per bacterial inactivation curve are summarized in Table 3. The $C_{res}$ values were similar in all models, indicating this concentration may be a critical density for the bacterial population to persist. Nonetheless, these values were slightly above the limit of detection. Note, that in all our experiments the bacteria were no more detectable by plating at the end of the observation periods. A persisting population was only observed in the autoclaved, artificial microcosms (see S1 Fig). The $\alpha$ parameter ranges between 0.03 and 4.61 1/day, where a higher $\alpha$ reflects a faster decline in concentration over time. Moreover, $\alpha$ and $\beta$ both affect the bacterial decline. The shaping parameter $\beta$ ranges between 0.49 and 5.63 [–]. The fitting plotted in R5 and R6 (Fig 3) have similar values for $\alpha$ and $C_{res}$, but differ in $\beta$. In fact, R5 has the longest shoulder where the initial population stays constant for nearly 20 days ($\beta = 5.63$). In contrast, the shoulder in R6 is smaller ($\beta = 3.1$) and observed during the first 15 days. There is also an influence of temperature on $\alpha$ and $\beta$ in our experiments, as seen for example in the die-off of *P. carotovorum* sp. *carotovorum* in natural oxic TDW (see Table 3, P1 & P2). There, $\alpha$ is 0.15 at 10˚C while $\beta$ is 5.61 (P1). At 25˚C the die-off is two times faster resulting in a higher $\alpha$ of 0.39 and a lower $\beta$ of 2.68 producing a two times shorter shoulder phase than at 10˚C. We observed this relation of the shoulder phase duration in days and the parameter estimate also with the other two bacteria.

## 4 Discussion

During MAR for agriculture, infiltrated TDW may contain plant pathogens that can still be present in the abstracted water after aquifer storage. Therefore, the die-off of bacterial plant pathogens in the water phase as one of the crucial removal processes during MAR was evaluated.

### 4.1 Influence of temperature

Overall, *R. solanacearum* was more persistent in the natural water microcosms than the soft rot Pectobacteriaceae (SRP), and the pathogens were differently susceptible to the tested temperatures. Whereas *R. solanacearum* copes better with the warmer conditions in the natural oxic TDW, the SRP survive better in the 10˚C microcosms, also supported by the results of the filtered TDW. The temperature dependency on the culturability of *R. solanacearum* has been shown earlier where the bacterium was monitored in a Spanish river over a two year period. The bacterium was not detectable with cultivation based techniques during the colder winter months, but reappeared when the temperature rose above 14˚C [43]. The absence of *R. solanacearum* after the exposure to lower temperatures can be linked to die-off, but also to the viable but non-culturable (VBNC) state the bacterium can enter [44], which was not evaluated in this study. To escape from hostile and low water temperatures during winter time, the bacteria survive in host plants growing along water streams. In particular, *R. solanacearum* was isolated from the bittersweet plant *Solanum dulcarama* [32], or from the stinging nettle *Urtica dioica* [10], from where it can get released when the water temperature starts to increase.

### 4.2 Influence of oxygen

It becomes clear that other factors than temperature influenced the die-off of the bacteria by comparing the die-off periods in natural oxic TDW and anoxic AW, both performed at 10˚C. Surprisingly, the die-off was not faster in anoxic microcosms but was decelerated at least two times. Whereas the SRP are described as facultative anaerobic bacteria that can switch their metabolism to nitrate respiration [45], little is known about the anaerobic metabolism of *R.*

*solanacearum* as it is generally described as aerobic organism. Wakimoto et al. [46] mentioned that *R. solanacearum* stayed viable in sterile anoxic water without stating further details. More recently, Dalsing et al. [47] explored the nitrogen metabolism of *R. solanacearum* phylotype I demonstrating its ability to grow better under anoxic conditions if nitrate ($>$ 62 mg/L) was supplemented in a liquid broth medium compared to growth in the same broth without nitrate. To add a higher concentration of nitrate (e.g. 1.5 g/L, as tested in [47]) would be not representative for the concentrations found in our water types. In our study, the addition of 50 mg/L nitrate did not have a significant effect on the die-off of *R. solanacearum*, even though the AW has a very low nitrate concentration (0.4 mg/L). Nitrate could have increased the survival serving as terminal electron acceptor under the anoxic conditions. Therefore, our study is the first to analyze the die-off of *R. solanacearum* phylotype II under anoxic conditions in natural water. Further investigation should elucidate the metabolic activity of *R. solanacearum* under these conditions. *R. solanacearum* not only persisted under anoxic conditions, but was culturable for a prolonged period at 10°C in the anoxic AW. For plant pathogens, the adaptation to low oxygen is important. They can encounter low oxygen conditions in the environment, or during the invasion of the plant host while thriving in the plant xylem; a requisite for a successful infection [48, 49]. For example, the absence of oxygen induces virulence related genes in *D. solani* enhancing the chance of a successful plant invasion by the pathogen [50]. Additionally, anaerobic nitrate respiration by the formerly named *Erwinia carotovora* was activated in the absence of oxygen; conditions that the bacterium experiences during potato tuber invasion [45].

## 4.3 Influence of microbiota

Temperature not only influences the pathogens themselves, but increases biotic interactions of the native microbiota. As reported by Álvarez et al. [22], the die-off of *R. solanacearum* in natural river microcosms was much faster at 24°C (4 days) than at 14°C (28 days), which the authors explained with increased microbial activity at the higher temperature, that negatively affected the pathogen. In contrast to this study, we found in our experiments that *R. solanacearum* was longer culturable in natural TDW (about 20 days) at 25 than at 10°C. Whereas for the SRP, we observed a faster decline within 4 days at a higher temperature (25°C) and a better persistence at the lower temperature in natural TDW. Nevertheless, the temperature effect on the die-off of the SRP was negligible in autoclaved TDW microcosms, when comparing the inactivation curves at 10 and 25°C (see S1 Fig). The long-term persistence in sterile aquatic microcosms, where the indigenous microbiota was absent, has been shown earlier for the SRP [25, 51], as well as for *R. solanacearum* [22–24]. In addition, our results demonstrated that *D. solani* can persist in sterile autoclaved water for a prolonged time which seem to contrast with the findings of van Doorn et al. [25] where *Dickeya* spp. rendered unculturable after a few days of incubation in sterile basin or ditchwater. *R. solanacearum* persisted two to three times longer in natural TDW at 10 and 25°C than the SRP, which indicates that the former is better adapted to the conditions in the natural TDW and against increased biotic interactions at 25°C. In the anoxic AW, the addition of nitrate to the natural AW microcosms might have also increased biotic interactions by stimulating the growth of denitrifies as part of the aquifer microbial community. This could explain the faster die-off of *D. solani* in AW supplemented with 50 mg/L nitrate. On the contrary, the die-off of *R. solanacearum* was not influenced by nitrate addition and did not shorten its survival, supporting its better adaptation against the existing microbiota. In this study, the indigenous microbiota was only represented by the culturable community (growing in TSA and R2A, respectively). Further research is required to elucidate any specific interactions between the examined pathogens and their environment

including other microorganisms in the microbial community. For example, Lowe-Power et al. [49] outlined that low cell densities of *R. solanacearum* sensed via quorum sensing, induce a metabolic strategy that allows the pathogen to exploit a broader variety of nutrients. This increases the pathogen's fitness and competition against other microbes, justifying the long tail in the die-off curves of *R. solanacearum* where the bacteria persist over an extended period at low concentrations. Interestingly, we observed similar and longer die-off periods in anoxic AW than in TDW for all the tested pathogens. Although TDW and AW presented comparable concentrations of culturable bacteria the composition of the microbiota might differ between both, as previously reported [52]. These authors stated that the total diversity of the microbiota in shallow aquifers is lower than in the overlying surface, which could support the different die-off periods of the pathogens in natural TDW and AW, observed in the present study. In fact, the TDW used in our microcosms is rain water that passes through the upper nutrient rich soil rhizosphere, where microorganisms such as protozoa, bacteria and viruses can detach from the soil surfaces and end up in the TDW used for infiltration. This can result in a heterogeneous microbial community and its composition is dynamic depending on the seasonal variation. Although, the effects of the different fractions of the native microbiota on the pathogens were not accessed in this study, previous studies reported that the native microorganisms (e.g., protozoa, bacteria and bacteriophages) increased the inactivation of *R. solanacearum* significantly [22]. Protozoa play an important role in the interaction with the pathogens as they can reduce bacterial populations by grazing [53]. They are aerobic organisms, thus, they will not be found as part of the active indigenous microbial community in the anoxic AW environment. In our study, the absence of protozoa in anoxic AW could be one factor influencing the prolonged persistence of the pathogens, under this condition. Furthermore, the pathogens themselves actively can increase their survival chance by producing enzymes and virulence factors, as reported for *D. solani*. These metabolic changes are relevant for the adaptation to unfavorable environments and to improve the competition against other microorganisms [54]. The obtained data suggests that the inoculated pathogens competed better against the AW microbiota than to the TDW, where fewer predators (e.g. protozoa) are present, or where the absence of oxygen induces metabolic responses, increasing the pathogen fitness in the AW. This study focused on the die-off kinetics of the pathogens and only analyzed the culturable part of the indigenous microbiota. Interactions between the pathogens and specific microorganisms or bacteriophages may affect the die-off kinetics and needs to be further addressed.

### 4.4 Model

The obtained inactivation curves of *R. solanacearum* in natural TDW are comparable with die-off periods from an earlier study in natural drainage water, where *R. solanacearum* was not culturable after 16 days at 12°C, but still detectable (4 CFU/mL) at 20°C after 32 days; both experiments by van Elsas et al. used an inoculation concentration of $5 \times 10^3$ CFU/mL [23]. The comparison of the *R. solanacearum* die-off curves with ours reveal similar features regarding the curve shape: initially, the die-off of *R. solanacearum* in natural oxic drainage water is fast and without a distinct shoulder, followed by a slower die-off until the bacterial population remains stable at a low concentration, and until the bacteria are no more culturable (Fig 3, R1 +2). Although the die-off curves in both studies depict clearly a non-linear trend, the authors [23] only reported first order die-off rates. As a rough estimation of die-off times this might be sufficient, but it could underestimate the pathogen's survival in the environment, if applied in risk modelling and if potentially contaminated water is recycled as irrigation water. Therefore, we propose the use of the non-linear Weibull + tail model which includes parameters to better describe the die-off pattern of the bacterial plant pathogens, as it accounts for convex or

concave ($\beta$ parameter), as well as tail-shaped curve shape ($C_{res}$). The model has the advantage that it incorporates the simpler Weibull model and a log-linear model. As a result, the flexibility of the model allowed to fit 17 different datasets. Nevertheless, it is not always possible to compare the resulting parameter estimates of two different bacteria. For example, $\beta$ was similar (~ 5.5) in two cases: the die-off of *D. solani* in filtered TDW at 10˚C, representing a shoulder duration of around 35 days; and the die-off of *R. solanacearum* in anoxic AW, where the shoulder phase was observed for at least 15 days. Consequently, some general assumptions can be made evaluating the parameter estimates, but a direct comparison of the parameters between the bacterial species is not possible. On the one hand, this can be explained by the Weibull model equation itself (see model 1 or 2), where both $\alpha$ and $\beta$ influence the decay time as the two parameters are inversely related: if a large $\beta$ is needed to describe the shoulder, it means that $\alpha$ must become lower as otherwise the effective rate constant gets too high. On the other hand, it is not possible to compare the parameter estimates of different bacteria as each species has different metabolic stress responses, and thereby different lethal times. The Weibull model takes into account this heterogeneity within a bacterial population, assuming that the cells are not equally resistant to the environmental stresses they are confronted with [26]. The existence of subpopulations has been described earlier. van Elsas et al. [23] characterized morphological differences of a *R. solanacearum* population. Along with these results, Álvarez et al. [27] investigated the different survival strategies of *R. solanacearum*, that included the formation of viable but not culturable (VBNC) state, persister cells, or the change of their cell shape from bacilli to coccoid form that also alters their cell metabolic activities.

The $C_{res}$ parameter which accounts for a persisting population is another important criterion for the model selection. The Weibull model considers the distribution of different stress tolerances within the population, but does not describe the mechanisms of the resistances as described in the previous paragraph [55]. Even though the $C_{res}$ parameter improves the prediction of the bacteria's die-off, if a persistent population is present, it also has a limitation. According to the Weibull + tail model, the bacterial concentration might never reach zero because it includes $C_{res}$. Therefore, the $C_{res}$ parameter should not be handled as an absolute value, but as a fraction of the initial $C_0$ concentration in risk modelling. Otherwise, the final concentration after treatment of even low (< 10 CFU/mL) contaminated water would mathematically always result in $C_{res}$ regardless of the initial $C_0$. However, under natural conditions, over much longer time frames, the populations get extinct and are no more present, which in our study was reached with the limit of culturability in all natural microcosm conditions. In our case, the prediction beyond the data cannot be executed with sufficient certainty and therefore, the situation where the bacteria get extinct cannot be reproduced with the Weibull + tail model [28], but would require the addition of another parameter to the formula. The use of different detection methods that are able to monitor lower concentrations could circumvent this problem or just postpone it to a later time point. When comparing with other studies that studied the die-off of *R. solanacearum* in natural waters [22, 23], a higher inoculation concentration ($10^6$ CFU/mL) did not increase the final concentration of the resistant population but increased the length of the die-off period. The awareness that such a persistent population exists, is especially important in the risk assessment as these bacteria may still cause disease. Die-off studies with human pathogens in water microcosms also illustrate that first-order inactivation does not sufficiently describe the pathogen's die-off because after an initially rapid inactivation, the pathogens persist at low concentrations for a prolonged period [56, 57]. Starved *R. solanacearum* populations from oligotrophic and sterile microcosms, were still able to cause disease after a prolonged period [27]. Nevertheless, as these experiments have been conducted under rather artificial conditions and used the highly susceptible tomato plant as

host, it makes it difficult to adopt them to natural environments. The $C_{res}$ parameter can serve as a worst case scenario where a low bacterial populations survives in the water.

However, the die-off in the water phase is only one removal process during MAR and other die-off processes, such as soil attachment, need to be taken into account within the risk assessment. If bacteria are still present after the MAR treatment, their concentrations will be low and future research needs to address the pathogenicity of low-inoculum concentrations. In our study, we have shown that the three economically relevant bacterial plant pathogens *R. solanacearum*, *D. solani* and *P. carotovorum* sp. *carotovorum* can persist in natural waters days to weeks, thereby posing a risk if recycling water systems are applied for irrigation without further treatment or insufficient residence times. Our non-linear models describe well the progression of the bacteria's die-off in natural water environments and could be applied to other human or plant pathogens. The results will contribute to define the operation of a MAR system in order to provide safe irrigation water, where bacterial concentrations are so low that there is no risk inducing plant diseases.

## 5 Conclusions

In this research we determined the die-off kinetics of three bacterial plant pathogens (*Ralstonia solanacearum*, *Pectobacterium carotovorum* sp. *carotovorum*, and *Dickeya solani*) in natural oxic TDW and in natural anoxic groundwater below an agricultural field. The decline in bacterial concentration by 3-$\log_{10}$ occurred between 6 to 50 days and was faster in oxic tile drainage water than in anoxic water from the aquifer. As a result, a variety of die-off curves with different shapes was obtained and we developed a flexible non-linear Weibull model that allowed to model the different bacterial die-off curves. These models are needed to reliably predict the die-off of relevant pathogens in aqueous environments and specifically in MAR systems. In the future, water reuse schemes will gain more importance as fresh water scarcity increases which will affect especially agricultural production that accounts for about 70% of fresh water use. MAR is an option to store fresh TDW in the subsurface for later use as irrigation water but water quality, particularly, related to the presence of plant pathogens needs to be considered. Our results will be implemented in microbial risk assessments setting guidelines for a safe application of water reuse schemes such as MAR. This study contributes to the knowledge about the survival of plant pathogens in the agro-ecosystem which is crucial for plant disease management.

## Supporting information

**S1 Fig. Die-off in autoclaved TDW.** Die-off of *Dickeya solani* (D), *Pectobacterium carotovorum* sp. *carotovorum* (P), and *Ralstonia solanacearum* (R) in microcosms in autoclaved TDW at two temperatures (10°C, first row, and 25°C, second row), shown as $\log_{10}$ [CFU/mL] vs. time [days]. Points represent the plate counts in duplicate of two microcosms per treatment.
(TIFF)

**S1 Table. Enumeration of culturable bacteria in natural waters from a managed aquifer recharge system.**
(DOCX)

**S2 Table. Akaike information criterion (AIC) values; the minimum value among the three models is with a colored background and depicts the model with the best fit.**
(DOCX)

**S1 File. R script for non-linear modelling.**
(DOCX)

**S2 File. Experimental data to model die-off.**
(PDF)

## Acknowledgments

We like to thank Alexandre Jousset (Department Ecology and Biodiversity, University of Utrecht) to facilitate the work with *R. solanacearum* in the quarantine laboratory. We thank Clara Arderiu Carné for the assistance during the experiment with autoclaved microcosms and preliminary experiments for anoxic microcosms as part of her Erasmus+ internship.

## Author Contributions

**Conceptualization:** Boris M. van Breukelen, Gertjan Medema, Jouke Velstra.

**Data curation:** Carina Eisfeld, Jan M. van der Wolf, Jack F. Schijven.

**Formal analysis:** Carina Eisfeld, Jack F. Schijven.

**Funding acquisition:** Boris M. van Breukelen, Gertjan Medema, Jouke Velstra, Jack F. Schijven.

**Investigation:** Carina Eisfeld.

**Methodology:** Carina Eisfeld, Jan M. van der Wolf, Gertjan Medema, Jack F. Schijven.

**Project administration:** Jan M. van der Wolf, Boris M. van Breukelen, Jack F. Schijven.

**Resources:** Jan M. van der Wolf.

**Supervision:** Jan M. van der Wolf, Boris M. van Breukelen, Gertjan Medema, Jack F. Schijven.

**Validation:** Jan M. van der Wolf.

**Visualization:** Carina Eisfeld, Jack F. Schijven.

**Writing – original draft:** Carina Eisfeld, Jan M. van der Wolf, Boris M. van Breukelen, Jack F. Schijven.

**Writing – review & editing:** Carina Eisfeld, Jan M. van der Wolf, Boris M. van Breukelen, Jack F. Schijven.

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
