## [Decision Letter · Decision Letter 0]

21 Dec 2020

PONE-D-20-35253

Die-off of plant pathogenic bacteria in tile drainage and anoxic water from a managed aquifer recharge site

PLOS ONE

Dear Dr. Eisfeld,

Thank you for submitting your manuscript to PLOS ONE. After careful consideration, we feel that it has merit but does not fully meet PLOS ONE’s publication criteria as it currently stands. Therefore, we invite you to submit a revised version of the manuscript that addresses the points raised during the review process.

In particular, criteria 3 and 4 (https://journals.plos.org/plosone/s/criteria-for-publication) have not been fully achieved. Both reviewers addressed the need to clarify experimental procedures, particularly related to the autoclaved dataset. The criterium to exclude this dataset is not clear, the explanation given in lines 345-347 “In the experiments with autoclaved water, in the initial stage, growth of the pathogens was observed. These data were not modelled, because autoclaved water is not representative of the actual natural conditions during MAR.” is vague. Why is the filtered TDW but not autoclaved TDW representative of the natural conditions? Did authors measure the total and dissolved organic carbon in the natural and treated waters? Carbon content, particularly, dissolved organic carbon, would be an important parameter to consider in the die off of the bacteria, being a parameter affected particularly in the autoclaved treatment.

Additionally, in the description of the treatments (line 212-226), it is not explained why the influence of the role of the natural microbial community in the anoxic natural aquifer was not evaluated. Why the authors did not consider it an important factor under these conditions?

As pointed by the reviewers, it should be clearly stated by the authors that within the bacterial community only the culturable part was determined in the present study.

In general, the English is good, the discussion is well written, but the abstract and results section needs to be improved. The text needs to be fluid and the abstract needs to be more concise and more assertive, so I advise a careful edition by someone whose first language is English.

I also have few additional points:

Lines 64-66, please improve these sentences, and be careful when referring “we investigate the microbiological aspects of MAR”, as only the bacterial community was determined, and within this only the culturable members were determined;

Line 95, replace “D. solani and concentrations” by “D. solani at concentrations”

Line 182, replace “Excess rainwater rain water reaches” by “Excess rainwater reaches”;

Line 361-362, improve the sentence;

Line 373-375, improve the sentence, is not clear;

Line 400, “In addition, under anaerobic conditions the tailing effect is more pronounced (Fig 3, R5 & R6).”, is it? It seem that a longer tail is visible in R4 (filtered TDW, 25 ºC);

Line 402-406, these two sentences can be combined, in a simpler sentence;

Line 440, replace “Graph R5…” by “The fitting plotted in R5 and R6”;

Line 441-443, replace “In the graph, R5 has the longest shoulder (β = 5.63) where the initial population stays constant for nearly 20 days. In contrast, the shoulder in R6 is observed during the first 15 days expressed in a smaller β of 3.1.” by “In fact, R5 has the longest shoulder where the initial population stays constant for nearly 20 days (β = 5.63). In contrast, the shoulder in R6 is a smaller (β = 3.1).”;

Line 554, replace “As a result, this flexibility” by “As a result, the flexibility”.

We look forward to receiving your revised manuscript.

Kind regards,

Ana R. Lopes, PhD

Academic Editor

PLOS ONE

Journal Requirements:

2. Thank you for stating the following in the Competing Interests/Financial Disclosure* (delete as necessary) section:

"This research has been financially supported by the Netherlands Organisation for Scientific Research (NWO; Topsector Water Call 2016; project acronym AGRIMAR; contract number: ALWTW.2016.023) with co-funding from private partners Acacia Water B.V., Broere Beregening B.V., and Delphy B.V. The funders had no role in study design, data collection and analysis, decision to publish, or preparation of the manuscript."

We note that one or more of the authors are employed by a commercial company: Acacia water B.V.

2.1. Please provide an amended Funding Statement declaring this commercial affiliation, as well as a statement regarding the Role of Funders in your study. If the funding organization did not play a role in the study design, data collection and analysis, decision to publish, or preparation of the manuscript and only provided financial support in the form of authors' salaries and/or research materials, please review your statements relating to the author contributions, and ensure you have specifically and accurately indicated the role(s) that these authors had in your study. You can update author roles in the Author Contributions section of the online submission form.

2.2. Please also provide an updated Competing Interests Statement declaring this commercial affiliation along with any other relevant declarations relating to employment, consultancy, patents, products in development, or marketed products, etc.  

Reviewers' comments:

Reviewer's Responses to Questions

**Comments to the Author**

1. Is the manuscript technically sound, and do the data support the conclusions?

Reviewer #1: Yes

Reviewer #2: Yes

2. Has the statistical analysis been performed appropriately and rigorously? 

Reviewer #1: Yes

Reviewer #2: Yes

3. Have the authors made all data underlying the findings in their manuscript fully available?

Reviewer #1: Yes

Reviewer #2: Yes

4. Is the manuscript presented in an intelligible fashion and written in standard English?

Reviewer #1: Yes

Reviewer #2: Yes

5. Review Comments to the Author

Reviewer #1: Summary

With increasing concerns for shortage of fresh water for agriculture, managed aquifer recharge (MAR) practice that stores excess fresh water, such as tile drainage water (TDW), offers a valid source of irrigation water for crops. Unfortunately, the water can be contaminated and be responsible for dissemination of plant pathogens upon irrigation. In order to understand and develop a model to estimate the survival and persistence of plant pathogens in TDW and AW (aquifer water), the authors determined the die-off rate of three plant pathogens as a function of oxygen, temperature, and resident microbiota. They developed a flexible non-linear Weibull model to predict bacterial die-off to determine the persistence of the plant pathogens in TDW and AW. Although each bacterial pathogen behaves differently, the developed model can be used as a part of an initial pathogen risk assessment to determine the potential safety of the stored water for safe irrigation of agricultural crop. It is hoped that with the model, they can develop a safe approach to remove the potential pathogens from the stored water.

Although there are some concerns with the experimental procedures, the authors were largely successful in developing a non-linear model to predict the survivability and persistence of three plant pathogens as a function of oxygen, temperature, and the resident microbiota.

Concerns

Most of the major concerns were with microbiological experiments.

1. How were the indigenous microbiota determined? Based on the description on page 8, lines 193-194, it is unclear whether TSA or R2A agar were used for both oxic and anoxic water samples and how the plates were incubated. One would hope that the anoxic water sample containing plates were incubated under anaerobic conditions in order to determine the anaerobes present in the sample but it is not clear from their description. Furthermore, the incubation temperature is also important to grow organisms that have adapted to colder temperature. Another important aspect that should have been taken into account for both indigenous microbes as well as inoculated plant pathogens after a prolonged incubation in the water samples is to have plated them on nutrient poor media. The authors used R2A but it is not clear from their description, exactly which samples were plated on TSA versus R2A. Finally, given that less than 1% of the bacteria in nature can be cultured, the authors SHOULD have performed metagenomic analysis to determine the indigenous microbiota in these water samples because they discuss potential competition by these organisms on plant pathogens throughout the manuscript. A better understanding of the indigenous microbiota and their effect on pathogens will also help understand the pathogen persistence in water from different locations which would have to be performed to validate the model.

2. Since the authors did not use the autoclaved water data in their modeling because of the initial growth of the inoculated bacteria (likely due to the dead indigenous organisms serving as nutrients), it is unclear whether discussion of this condition serves any purpose for this manuscript, even as supplemental data.

3. Although it may be out of scope for this particular manuscript, it would have been helpful to have addressed the relative metabolic change of the three bacteria as a function of temperature and oxygen. That may help explain the observed data especially with regards to the temperature which the authors call "temperature sensitivity" on page 23, line 394.

4. Figure 3 needs to be improved to clearly show the data. Unlike other figures in the manuscript, this figure was fuzzy and out of focus.

5. There were some spelling errors that should be corrected.

6. This is not a concern but a positive critique. The Discussion was well-written and addressed most of the concerns of the data in the manuscript with the exception of those addressed above.

Reviewer #2: GENERAL COMMENTS:

This manuscript investigated the die-off of three important plant pathogens, Ralstonia solanocearum, Dickeya solani, and Pectobacterium carotovorum sp. carotovorum in natural tile drainage water and anoxic aquifer water under different conditions. The collected data was used to fit the best die-off model by comparing a non-linear Weibull+ tail, a Weibull, and a modified log-linear model in order to predict the fate of these pathogens for risk assessments in recycling water systems. This manuscript is worthy of publication in PLOS One following major revision in terms of its applicable value. However, there are issues in the manuscript that need to be clarified and revised before publishing.

Suggestions for improvement of the manuscript are outlined below.

1. Pg2, Ln35, Ln82: change ‘Pectobacteriaceae’ to ‘Pectobacteriaceae’. Family name should not be italic. This occurred in several places across the manuscript, so please address it.

2. Pg2, Ln64-66: consider deleting the two sentences, or move them to the last paragraph of the introduction.

3. Pg3, Ln68: consider adding ‘also known as brown rot’ after ‘They can cause bacterial wilt’. Your audience (without a background in plant pathology) may be confused when you mention potato brown rot in Ln72.

4. Pg4, Ln85-86: suggest deleting “For example… glasshouse ornamental crops”. This is a redundant sentence. You have mentioned its broad host range in Ln69.

5. Pg4, Ln87-97: Please consider merging this paragraph with the previous paragraph.

6. Pg4, Ln93: Change ‘(e.g. [16])’ to ‘[16]’.

7. Pg4, Ln94-95: consider changing to ‘Kastelein et al. [17] reported that spray inoculation of potato leaves with 102 CFU/mL D. solani resulted in diseased plants in greenhouse experiments.’

8. Pg4, Ln95-97: consider deleting ‘Moreover… potato material’. You have mentioned SRP can be found in the environment in Ln91.

9. Pg5, Ln117: change ‘, as well as the natural microbiota,’ to ‘and the natural microbiota’.

10: Pg5, Ln119: change ‘No study, however, was found that describes’ to ‘However, no study has reported that the die-off…’.

11: Pg6, Ln154: change ‘Streptomycin’ to ‘streptomycin’.

12. Pg7, Ln 163-164: change ‘the aquifer storage… (ASTR) system’ to ‘The ASTR system’; change ‘tile drainage water (TDW)’ to ‘TDW’. This also applies to other places across the manuscript.

13. Pg7, Ln166: delete ‘Figure 1 shows a scheme of the ASTR system’, and add ‘, as shown in Figure 1,’ right after ‘The ASTR system’ in Ln164.

14: Pg8, Ln182: delete ‘rain water’.

15. Pg8, Ln186: add ‘.’ after ‘irrigation’.

16. Pg8, Ln193-194: please clarify this sentence. You can only assess culturable microbes by plating on medium. The plating method cannot represent the microbiota. This issue occurred in several places across the manuscript. Please address them as well. In addition, please spell out R2A agar as this is the first time it appeared in the text.

17. Pg11, Ln256-258: rewrite these two sentences and add references.

18. Pg11, Ln261: change ‘, plus’ to ‘and’.

19. Pg11, ln268: spell out SMSA.

20. Pg11, Ln269, Ln271: change ‘Rifampicin’ to ‘rifampicin’; change ‘Streptomycin’ to ‘streptomycin’. This occurred in several other places, please address them in order to be consistent.

21. Pg11-12, Ln272-274: add reference(s).

22. Pg12, Ln 274-276: Again, you can only determine culturable bacteria through TSA or R2A. It is not correct to use the term ‘microbiota’ since you only measured culturable bacteria.

23. Pg12, Ln278: how many replicates?

24. Pg12, Ln283L: does ‘the bacterial concentration’ here refer to the culturable bacteria or the tested three phytopathogens?

25. Pg13, Ln318-320: suggest changing to ‘All three models were applied to fit the experimental data and the best model was chosen based on the Akaike Information Criterion (AIC)’.

26. Pg13, Ln320-325: suggest deleting the description on AIC (this is commonly known, and you have directed your audience to the reference as well).

27. Pg15, Ln339: change ‘microbiota’ to ‘culturable bacterial’.

28. Pg15, Ln340: what is the SD of the background bacterial concentration (102 -104 CFU/mL)?

29. Pg15, Ln350: delete the comma between ‘modelling’ and ‘because’. This also occurred in other places, please address them.

30. Pg16, Ln366: add ‘both’ before ‘selected’.

31. Pg16, Ln370-371 ‘The results of experiments… in S1 Fig’, please at least briefly describe your results of the autoclaved experiments here.

32: Pg16, Ln377: add ‘was’ before ‘slower’.

33: Pg18-19, ln439-440: delete ‘, which will be discussed later.’

34: Pg21, Ln465: delete ‘state’.

35. Pg21, Ln471: delete ‘,’.

36: Pg22, Ln493: delete ‘;’.

37: Pg22-23, session 4.3: Is there a difference between culturable bacteria in the background in between TDW and AW? Where is the data??

38: Pg24, Ln541: change ‘no more’ to ‘not’.

39: Pg25, Ln563-564: change ‘On the other hand’ to ‘In addition’.

40: Pg26, Ln 604: change ‘can be’ to ‘could be’

41. Pg26, Ln 606-607: reconsider the statement: ‘no risk inducing plant diseases.’ It may be the fact, but I would not say without a supporting data or listing a reference.

42. Pg28, reference session: font and size are different compared to previous sessions.

6. PLOS authors have the option to publish the peer review history of their article (what does this mean?). If published, this will include your full peer review and any attached files.

Reviewer #1: No

Reviewer #2: No

---

## [Author Response · Author response to Decision Letter 0]

5 Feb 2021

Please also find our comments in the submitted document "Response to the reviewers". 

Reply to the Editor: 

Thank you for handling our manuscript and considering it for publication in PLOS ONE. 

In the following, we address all comments raised by you and the reviewers and explain how we improved our paper to meet the publication criteria of PLOS ONE. Replies by the authors are indicated in pink, small textural changes are edited in the manuscript and are declared with “Done.” in this rebuttal letter. All changes can be followed up in the revised manuscript with track changes which was uploaded together with this rebuttal letter. Moreover, we adapted the funding statement according to PLOS ONE. The author with commercial affiliation was involved in the conceptualization and funding acquisition, but the funding did not influence the study design or any results. 

Specific comments by the Editor: 

1. In particular, criteria 3 and 4 (https://journals.plos.org/plosone/s/criteria-for-publication) have not been fully achieved. Both reviewers addressed the need to clarify experimental procedures, particularly related to the autoclaved dataset. The criterium to exclude this dataset is not clear, the explanation given in lines 345-347 “In the experiments with autoclaved water, in the initial stage, growth of the pathogens was observed. These data were not modelled, because autoclaved water is not representative of the actual natural conditions during MAR.” is vague. Why is the filtered TDW but not autoclaved TDW representative of the natural conditions? 

Reply by the authors: We wanted to measure and simulate the die-off under natural conditions and know the effects of the naturally present microbiota. To investigate the latter we applied filtration or autoclaving, both widely used methods to remove/inactivate the naturally present microbiota. Filtration does not change the abiotic conditions whereas autoclaving does change the chemical composition. We observed temporal growth of the bacteria we added only under these conditions and hypothesized that this is likely due to the thermal breakdown of organic compounds in the autoclave, which made these compounds more readily biodegradable. As autoclaving does not represent natural abiotic conditions in comparison to the filtered microcosms, we did not model those results. In addition, the autoclaved results could demonstrate a new finding as Dickeya solani persisted in sterile autoclaved water for a prolonged time. This contrasts with the results of van Doorn et. al. (2008) who analyzed the survival of Dickeya spp. in sterile waters and reported that the bacteria rendered unculturable after a few days of incubation. To point this out, we added the following sentence after line 505 of the original manuscript: 

“In addition, our results demonstrated that D. solani can persist in sterile autoclaved water for a prolonged time which seem to contrast with the findings of van Doorn et.al. [26] where Dickeya spp. rendered unculturable after a few days of incubation in sterile basin or ditchwater.”

Eventually, the autoclaved data serve as a control and we used them to compare results of natural and filtered treatments (line numbers refer to the original manuscript): 

- line 396 et. seq.: “A similar observation was made in autoclaved TDW at 10 °C, where R. solanacearum is no longer culturable after a period of 30 days, even though the bacteria were culturable for a long period of 150 days at 25 °C (see S1 Fig).”

- line 421 et. seq.: “The faster die-off at 25 °C compared to 10 °C was noticeable in untreated and in filtered TDW, implying lower resistance of the SRP at a higher temperature. Still, the die-off in autoclaved TDW was similar at both temperatures.”

- line 436 et. seq.: “Note, that in all our experiments the bacteria were no more detectable by plating at the end of the observation periods. A persisting population was only observed in the autoclaved, artificial microcosms (see S1 Fig).”

The text passage in line 345-347 et. seq. (original manuscript) has been modified accordingly: 

“Long-term persistence of the pathogens was demonstrated in the autoclaved microcosms where optimal conditions (available nutrients, no competition with other microorganism) resulted in growth of the pathogens in the beginning of the experiments. In the autoclaved microcosms the physicochemical composition of the water is changed during the heating process. In contrast, the filtration only removes the biotic fraction from the water leaves the abiotic conditions unchanged. This allows to study the influence of the biotic factors in oxic microcosms when comparing natural and 0.22 µm filtered TDW.”

2. Did authors measure the total and dissolved organic carbon in the natural and treated waters? Carbon content, particularly, dissolved organic carbon, would be an important parameter to consider in the die off of the bacteria, being a parameter affected particularly in the autoclaved treatment.

Reply by the authors: We did not measure total or dissolved organic carbon, but measured the chemical oxygen demand which represents the overall organic compounds in the water. Note, that COD also measures NH4, which was zero in the tested water samples (see Table 2 of the manuscript). The following sentence was added in the results section (line 352 of the original manuscript) regarding COD: 

“The measured values of the chemical oxygen demand (COD) represent the total organic compounds in the water and they are representative for the range found in unpolluted surface waters (around 20 mg/L).”

3. Additionally, in the description of the treatments (line 212-226), it is not explained why the influence of the role of the natural microbial community in the anoxic natural aquifer was not evaluated. Why the authors did not consider it an important factor under these conditions?

Reply by the authors: The influence of the biotic fraction on the die-off of our target pathogens was shown under oxic conditions and therefore we considered it less relevant to demonstrate it also for anoxic water. 

4. As pointed by the reviewers, it should be clearly stated by the authors that within the bacterial community only the culturable part was determined in the present study. 

Reply by the authors: 

We thank the editor and reviewers for their critical comments regarding this topic. We removed the sentence in line 193-194 (original manuscript) and added the missing information regarding the enumeration of indigenous culturable bacteria to a) 2.3 of material and methods (starting from line 219 of the original manuscript) and b) 2.4 of material and methods (line 274 of the original manuscript): 

a)” For both the natural microcosms of oxic TDW and anoxic AW, a respective control microcosm that was not inoculated with pathogens was prepared.”

b) “The initial concentration of background culturable bacteria in the natural oxic and anoxic water samples was assessed by plating 0.1 mL on non-selective TSA and low-nutrient Reasoner’s 2A (R2A) agar (Oxoid; Thermo Fisher Scientific) and incubated at 25 °C. Anaerobic water samples were incubated anaerobically 25 °C using Oxoid AnaeroGen sachets (Oxoid; Thermo Fisher Scientific) to create an anaerobic atmosphere in a closed container. Additionally, total numbers of culturable bacteria were enumerated in all oxic TDW microcosms from both temperatures at the end of the die off period of the SRP.”

To be more specific on that topic, we added the following part to the results section in 3.1 (starting in line 339 of the original manuscript) and show the enumeration of culturable bacteria on TSA and R2A in S1 Table which we will add to the supplementary material.

 “In natural TDW at 10 and 25 °C, the background culturable bacterial levels in the control microcosms were 102 CFU/mL grown on TSA and R2A, and reached 9.6 – 9.8x103 CFU/mL after 16 days, grown on TSA or R2A respectively. In the control microcosms of anoxic AW with or without nitrate, the concentration of culturable indigenous bacteria was about 2.3x102 – 4x103 CFU/mL, grown on TSA or R2A respectively. S1 Table shows an overview of the enumeration of culturable microorganism on TSA and R2A present in the native TDW and AW.”

Please note that S1 Table (containing information about the AIC values) provided in the initial submission will be renumbered to S2 Table. 

5. Lines 64-66, please improve these sentences, and be careful when referring “we investigate the microbiological aspects of MAR”, as only the bacterial community was determined, and within this only the culturable members were determined;

Reply by the authors: The sentence in the introduction in lines 64-66 was modified: “In this study, we investigate the die off of bacterial plant pathogens in different water types from a MAR site to mimic the injection of contaminated water into the MAR system.” 

6. In general, the English is good, the discussion is well written, but the abstract and results section needs to be improved. The text needs to be fluid and the abstract needs to be more concise and more assertive, so I advise a careful edition by someone whose first language is English.

Reply by the authors: We thank the editor for the positive comment about the discussion. The manuscript has been examined by a professional English editor. We shortened the abstract, specifically the first lines, and modified passages in the results section to be more concise and to improve readability. To see all changes, we refer to the revised manuscript with track changes. 

7. Line 95, replace “D. solani and concentrations” by “D. solani at concentrations” Done. 

8. Line 182, replace “Excess rainwater rain water reaches” by “Excess rainwater reaches”; Done. 

9. Line 361-362, improve the sentence;

Reply by the authors: We made the following changes to clarify the sentence: “The experiments were concluded when the limit of detection of 3-10 CFU/mL was reached (visualized as horizontal dotted line in Figure 3) or the bacteria were no more detectable by plating (below the detection limit, as marked with an asterisk in Figure 3).”

10. Line 373-375, improve the sentence, is not clear;

Reply by the authors: We modified the sentence to make it more clear: “In anoxic AW, the addition of nitrate did not have a significant effect (P = 0.23) on the die-off of R. solanacearum.” 

11. Line 400, “In addition, under anaerobic conditions the tailing effect is more pronounced (Fig 3, R5 & R6).”, is it? It seem that a longer tail is visible in R4 (filtered TDW, 25 ºC);

Reply by the authors: We agree that the longest tailing effect of the inactivation curve can be observed in Graph R4 of Figure 3 as it was also described in the manuscript. To avoid any misunderstanding, we modified the sentence by combining the two sentences: 

“In filtered TDW at 25 °C, the bacteria persisted at a low concentration of ca. 10 CFU/mL for up to 60 days creating a long tail shape in the plotted curve (see Fig 3, R4), which can also be observed under anaerobic conditions for a period of ca. 25 days (Fig 3, R5 & R6). Surprisingly, the die-off of R. solanacearum under anoxic conditions took about 45 days and was two times slower than in natural TDW despite the low temperature of 10 °C and absence of oxygen.” 

12. Line 402-406, these two sentences can be combined, in a simpler sentence;

Reply by the authors: We changed the sentences to ease readability, see also comment 11: “Surprisingly, the die-off of R. solanacearum under anoxic conditions took about 45 days and was two times slower than in natural TDW regardless the low temperature of 10 °C and absence of oxygen.”

13. Line 440, replace “Graph R5…” by “The fitting plotted in R5 and R6”; Done.

14. Line 441-443, replace “In the graph, R5 has the longest shoulder (β = 5.63) where the initial population stays constant for nearly 20 days. In contrast, the shoulder in R6 is observed during the first 15 days expressed in a smaller β of 3.1.” by “ In fact, R5 has the longest shoulder where the initial population stays constant for nearly 20 days (β = 5.63). In contrast, the shoulder in R6 is a smaller (β = 3.1).”; Done. 

15. Line 554, replace “As a result, this flexibility” by “As a result, the flexibility”. Done. 

Additional modification of the manuscript by the authors: With outlook to future risk modelling, we added two sentences after line 578 (original manuscript) to avoid misunderstandings by the reader. 

 “Therefore, the Cres parameter should not be handled as an absolute value, but as a fraction of the initial C0 concentration in risk modelling. Otherwise, the final concentration after treatment of even low (> 10 CFU/mL) contaminated water would mathematically always result in Cres regardless of the initial C0.”

Reply to Reviewer 1: 

Reviewer #1: Summary

With increasing concerns for shortage of fresh water for agriculture, managed aquifer recharge (MAR) practice that stores excess fresh water, such as tile drainage water (TDW), offers a valid source of irrigation water for crops. Unfortunately, the water can be contaminated and be responsible for dissemination of plant pathogens upon irrigation. In order to understand and develop a model to estimate the survival and persistence of plant pathogens in TDW and AW (aquifer water), the authors determined the die-off rate of three plant pathogens as a function of oxygen, temperature, and resident microbiota. They developed a flexible non-linear Weibull model to predict bacterial die-off to determine the persistence of the plant pathogens in TDW and AW. Although each bacterial pathogen behaves differently, the developed model can be used as a part of an initial pathogen risk assessment to determine the potential safety of the stored water for safe irrigation of agricultural crop. It is hoped that with the model, they can develop a safe approach to remove the potential pathogens from the stored water.

Although there are some concerns with the experimental procedures, the authors were largely successful in developing a non-linear model to predict the survivability and persistence of three plant pathogens as a function of oxygen, temperature, and the resident microbiota.

Specific comments by reviewer 1: 

1. Most of the major concerns were with microbiological experiments.

How were the indigenous microbiota determined? Based on the description on page 8, lines 193-194, it is unclear whether TSA or R2A agar were used for both oxic and anoxic water samples and how the plates were incubated. One would hope that the anoxic water sample containing plates were incubated under anaerobic conditions in order to determine the anaerobes present in the sample but it is not clear from their description. Furthermore, the incubation temperature is also important to grow organisms that have adapted to colder temperature. Another important aspect that should have been taken into account for both indigenous microbes as well as inoculated plant pathogens after a prolonged incubation in the water samples is to have plated them on nutrient poor media. The authors used R2A but it is not clear from their description, exactly which samples were plated on TSA versus R2A.

Reply by the authors: We thank the reviewer for the critical comment to improve our manuscript and refer to editor’s comment #4 for the reply of the authors. 

2. Finally, given that less than 1% of the bacteria in nature can be cultured, the authors SHOULD have performed metagenomic analysis to determine the indigenous microbiota in these water samples because they discuss potential competition by these organisms on plant pathogens throughout the manuscript. A better understanding of the indigenous microbiota and their effect on pathogens will also help understand the pathogen persistence in water from different locations which would have to be performed to validate the model.

Reply by the authors: Metagenomic analysis could indeed give insight in to the microbiome in the water types, but not about their activity nor about direct interactions with the studied pathogens. We considered the application of metagenomics techniques, but decided that they would not have aided in explaining the established die off kinetics which is the scope of this paper. Further investigations on interactions between target pathogens and the microbiome (activity) are of interest but should be addressed in future research. 

To clarify that we only analyzed culturable microorganisms, we added the following sentences after line 512 (of the original manuscript) to the discussion for clarification: “In this study, the background microbiota was only characterized by plating TSA and R2A medium at 25 °C. Further research is required to elucidate any specific interactions between the examined pathogens and their environment including other microorganisms in the microbiome.” 

To point it out again, we also added a sentence at the end of 4.3 of the discussion (line 538 of the original manuscript): “This study focused on the die off kinetics of the pathogens and only analyzed the culturable part of the indigenous microbiota. Interactions between the pathogens and specific microorganisms or bacteriophages may affect the die-off kinetics, but this is topic for future research. “ 

3. Since the authors did not use the autoclaved water data in their modeling because of the initial growth of the inoculated bacteria (likely due to the dead indigenous organisms serving as nutrients), it is unclear whether discussion of this condition serves any purpose for this manuscript, even as supplemental data.

Reply by the authors: Thank you for the comment. Please see our reply on editor’s comment #1 regarding this topic. 

4. Although it may be out of scope for this particular manuscript, it would have been helpful to have addressed the relative metabolic change of the three bacteria as a function of temperature and oxygen. That may help explain the observed data especially with regards to the temperature which the authors call "temperature sensitivity" on page 23, line 394.

Reply by the authors: We agree with the reviewer, that the observation of the metabolic changes could have been an interesting aspect to observe. But as the reviewer already mentions, this was out of scope for this study. Moreover, it may be difficult to differentiate between the metabolic state of the target pathogen within a water microcosm where also other microorganism are present. 

5. Figure 3 needs to be improved to clearly show the data. Unlike other figures in the manuscript, this figure was fuzzy and out of focus. 

Reply by the authors: It was possible to download the figure separately. In the composed PDF, the figure is indeed blurry, but PLOS ONE offers the possibility to download the picture where it can be received with sufficient good quality (see also: https://www.editorialmanager.com/pone/download.aspx?id=27777079&guid=e3003e13-1a2e-4a23-b6c6-98cc59193cb6&scheme=1). 

6. There were some spelling errors that should be corrected.

7. This is not a concern but a positive critique. The Discussion was well-written and addressed most of the concerns of the data in the manuscript with the exception of those addressed above.

Specific comments by reviewer 2: 

Reviewer #2: GENERAL COMMENTS:

This manuscript investigated the die-off of three important plant pathogens, Ralstonia solanacearum, Dickeya solani, and Pectobacterium carotovorum sp. carotovorum in natural tile drainage water and anoxic aquifer water under different conditions. The collected data was used to fit the best die-off model by comparing a non-linear Weibull+ tail, a Weibull, and a modified log-linear model in order to predict the fate of these pathogens for risk assessments in recycling water systems. This manuscript is worthy of publication in PLOS One following major revision in terms of its applicable value. However, there are issues in the manuscript that need to be clarified and revised before publishing.

Reply by the authors: We thank the reviewer to consider our manuscript for publication in PLOS ONE and for the comments to improve our manuscript.

1. Suggestions for improvement of the manuscript are outlined below.

Pg2, Ln35, Ln82: change ‘Pectobacteriaceae’ to ‘Pectobacteriaceae’. Family name should not be italic. This occurred in several places across the manuscript, so please address it. Done. 

2. Pg2, Ln64-66: consider deleting the two sentences, or move them to the last paragraph of the introduction. Done, also according to editor, comment #5. 

3. Pg3, Ln68: consider adding ‘also known as brown rot’ after ‘They can cause bacterial wilt’. Your audience (without a background in plant pathology) may be confused when you mention potato brown rot in Ln72. Done.

4. Pg4, Ln85-86: suggest deleting “For example… glasshouse ornamental crops”. This is a redundant sentence. You have mentioned its broad host range in Ln69.

Reply by the authors: As this papers covers an interdisciplinary topic, we would like to keep this sentence to address a broader audience from different backgrounds. Readers with an engineering background who are not familiar with plant pathogens might not think about flower bulbs as hosts when initially talking about potato pathogens. 

5. Pg4, Ln87-97: Please consider merging this paragraph with the previous paragraph. Done. 

6. Pg4, Ln93: Change ‘(e.g. [16])’ to ‘[16]’. Done. 

7. Pg4, Ln94-95: consider changing to ‘Kastelein et al. [17] reported that spray inoculation of potato leaves with 102 CFU/mL D. solani resulted in diseased plants in greenhouse experiments.’ Done.

8. Pg4, Ln95-97: consider deleting ‘Moreover… potato material’. You have mentioned SRP can be found in the environment in Ln91. Done.

9. Pg5, Ln117: change ‘, as well as the natural microbiota,’ to ‘and the natural microbiota’. Done.

10. Pg5, Ln119: change ‘No study, however, was found that describes’ to ‘However, no study has reported that the die-off…’. Done.

11. Pg6, Ln154: change ‘Streptomycin’ to ‘streptomycin’. Done, also corrected in other parts of the manuscript, see also comment #20. 

12. Pg7, Ln 163-164: change ‘the aquifer storage… (ASTR) system’ to ‘The ASTR system’; change ‘tile drainage water (TDW)’ to ‘TDW’. This also applies to other places across the manuscript. Done.

13. Pg7, Ln166: delete ‘Figure 1 shows a scheme of the ASTR system’, and add ‘, as shown in Figure 1,’ right after ‘The ASTR system’ in Ln164. Done.

14. Pg8, Ln182: delete ‘rain water’. Done.

15. Pg8, Ln186: add ‘.’ after ‘irrigation’. Done. 

16. Pg8, Ln193-194: please clarify this sentence. You can only assess culturable microbes by plating on medium. The plating method cannot represent the microbiota. This issue occurred in several places across the manuscript. Please address them as well. In addition, please spell out R2A agar as this is the first time it appeared in the text.

Reply by the authors: Thank you for your comment. For the reply we refer to reviewer 1’s comment #1. 

17. Pg11, Ln256-258: rewrite these two sentences and add references.

Reply by the authors: The sentences have been modified and shortened and the reference placed accordingly: “Wenneker et. al. [10] detected R. solanacearum in ditches next to agricultural fields at concentrations ranging between 102 – 105 CFU/mL. Higher inoculum density may give the introduced pathogenic bacteria a competitive advantage against other microorganisms.”

18. Pg11, Ln261: change ‘, plus’ to ‘and’. Done.

19. Pg11, ln268: spell out SMSA. Done.

20. Pg11, Ln269, Ln271: change ‘Rifampicin’ to ‘rifampicin’; change ‘Streptomycin’ to ‘streptomycin’. This occurred in several other places, please address them in order to be consistent. Done, see also comment #11. 

21. Pg11-12, Ln272-274: add reference(s).

Reply by the authors: To select the best medium with the highest recovery rate and best suppression of other microorganisms, we did a small preliminary test, where a known concentration of bacteria was added to tile drainage water and recovered by plating. To clarify, we modified the sentence: 

“The selective media have been chosen as they showed a higher recovery rate from the environmental water samples in comparison to TSA supplemented with the respective antibiotic (unpublished results).”

22. Pg12, Ln 274-276: Again, you can only determine culturable bacteria through TSA or R2A. It is not correct to use the term ‘microbiota’ since you only measured culturable bacteria.

Reply by the authors: Thank you for the comment, we edited the sentence and replaced the term ‘microbiota by ‘culturable bacteria’. 

23. Pg12, Ln278: how many replicates? Done.

24. Pg12, Ln283L: does ‘the bacterial concentration’ here refer to the culturable bacteria or the tested three phytopathogens?

Reply by the authors: For clarification, we modified the sentence to: “… the bacterial concentration defined as culturable cells per mL… “

25. Pg13, Ln318-320: suggest changing to ‘All three models were applied to fit the experimental data and the best model was chosen based on the Akaike Information Criterion (AIC)’. Done.

26. Pg13, Ln320-325: suggest deleting the description on AIC (this is commonly known, and you have directed your audience to the reference as well).

Reply by the authors: We agree with the comment of the reviewer and shortened the paragraph about the AIC to the following: “The AIC considers the goodness of fit and parsimony of the models. For our experiments, the model with the smallest AIC value was therefore chosen as best model.”

27. Pg15, Ln339: change ‘microbiota’ to ‘culturable bacterial’. Done.

28. Pg15, Ln340: what is the SD of the background bacterial concentration (102 -104 CFU/mL)?

Reply by the authors: We added detailed information about the background bacterial concentrations in the S1 Table, which will be included to the supplementary material. We also refer to editor’s comment #4 and our reply on this topic. 

29. Pg15, Ln350: delete the comma between ‘modelling’ and ‘because’. This also occurred in other places, please address them. Done.

30. Pg16, Ln366: add ‘both’ before ‘selected’. Done.

31. Pg16, Ln370-371 ‘The results of experiments… in S1 Fig’, please at least briefly describe your results of the autoclaved experiments here.

Reply by the authors: We added a brief description after the sentence: “There, the bacteria started to grow at 10 and 25 °C reaching a concentration of 106 CFU/mL followed by a slow decline until they reached a concentration of at least 103 CFU/mL which lasted until the end of the observation period (150 days). The only exception from this trend was the die off of R. solanacearum in the autoclaved TDW at 10 °C which rendered non culturable after a period of approximately 30 days.”

32. Pg16, Ln377: add ‘was’ before ‘slower’. Done. 

33. Pg18-19, ln439-440: delete ‘, which will be discussed later.’ Done. 

34. Pg21, Ln465: delete ‘state’. Done.

35. Pg21, Ln471: delete ‘,’. Done.

36. Pg22, Ln493: delete ‘;’. Done.

37. Pg22-23, session 4.3: Is there a difference between culturable bacteria in the background in between TDW and AW? Where is the data?? Please see comment #28, the full data has now been added as S1 Table. 

38. Pg24, Ln541: change ‘no more’ to ‘not’. Done.

39. Pg25, Ln563-564: change ‘On the other hand’ to ‘In addition’. The sentence before, starts with ‘On the one hand’, hence the second expression ‘On the other hand’ needs to remain. 

40. Pg26, Ln 604: change ‘can be’ to ‘could be’ Done.

41. Pg26, Ln 606-607: reconsider the statement: ‘no risk inducing plant diseases.’ It may be the fact, but I would not say without a supporting data or listing a reference.

Reply by the authors: This comment is meant as a future outlook therefore the sentence contains the expression “in order to” and should not be treated as a final assumption. 

42. Pg28, reference session: font and size are different compared to previous sessions. The font and size of the references have been adapted to the rest of the manuscript.

---

## [Decision Letter · Decision Letter 1]

2 Mar 2021

PONE-D-20-35253R1

Die-off of plant pathogenic bacteria in tile drainage and anoxic water from a managed aquifer recharge site

PLOS ONE

Dear Dr. Eisfeld,

Thank you for submitting your manuscript to PLOS ONE. After careful consideration, we feel that it has merit but does not fully meet PLOS ONE’s publication criteria as it currently stands. Therefore, we invite you to submit a revised version of the manuscript that addresses the points raised during the review process.

Although the authors have addressed all the previous suggestions, there are still some points that need to be addressed. Regarding the culturable indigenous bacteria present in both oxic and anoxic microcosms, the inclusion of table S1 was important. But the duration of the experiments is not clear, based on the days presented in Fig 3 and on table S1. Enumerations in the controls are presented for day 0 and 16, but not all the experiments ended at day 16, right? Besides, for the inoculated microcosms there is no enumeration of the culturable bacteria at day 0, and also the enumeration at the end of the experiment is missing. The experiments were longer than 12 or 16 days, right?

Besides, in the material and methods the authors referred line 213-214 “Experiments with natural aquifer water where nitrate was added, were only tested with R. solanacearum and D. solani.”, why not evaluating the die-off of *P. carotovorum* sp. *carotovorum* under this condition?

Line 165-170, why is this information important?

Small issues that need to be addressed:

Abstract

Line 31 Delete “0.22 μm”;

Line 31 replace “anoxic aquifer water” by “anoxic aquifer water (AW)”;

Line 32 Replace “are” by “were”;

Line 40-41 remove “Higher temperature and the presence of background microbiota strongly

41 accelerated die-off.”;

Line 41, replace “In anoxic natural aquifer” by “Whereas In anoxic natural aquifer”;

Line 44-46, replace “The non-linear model provides a tool to reliably predict the die-off of plant pathogenic bacteria or other pathogenic microorganisms in the context of microbial risk assessment.”

by “The non-linear model was shown to be a reliable tool to predict the die-off of the analyzed plant pathogenic bacteria, suggesting its further application to other pathogenic microorganisms in the context of microbial risk assessment.”;

Material and Methods

Table 1, move the row “treatment” to the top of the table, above the row water type;

Line 209-211, replace “For both the natural microcosms of oxic TDW and anoxic AW, a respective control microcosm that was not inoculated with pathogens was prepared” by “For both the natural microcosms of oxic TDW and anoxic AW a non-inoculated control microcosm was prepared.

Line 214-218, replace “The temperatures were chosen as they represent, at the lower end, the nearly constant temperature in the aquifer (10 °C); while the upper end (25 °C) reflects infiltration of rainfall events during the warmer summer months. Moreover, it covers a temperature that is more representative for tropical regions, where these bacteria also play an important role as disease causing organisms.” By “The temperatures were chosen as they represent the nearly constant temperature in the aquifer (10 °C); while 25 °C reflects infiltration of rainfall events during the warmer summer months. Moreover, the last covers a temperature more representative of tropical regions, where these bacteria also play an important role as disease causing organisms.”;

Line 220, replace “microbiome” by “microbial”;

Line 221, remove “a sterile filter holder with receiver (Nalgene, USA) and”;

Line 265, replace “unpublished results” by “data not shown”;

Line 266/511/, replace “background” by “indigenous”;

Line 279, replace by “the bacterial” by “the pathogen”;

Figure 2, replace by “the population concentration” by “the pathogen concentration”;

Results

Line 363/367, replace “S1 Table” by “Table S2”;

Line 370, replace “in S1 Fig” by “Fig S1”;

Line 443, replace “faster resulting in a higher α of 0.39, and β is 2.68 producing a two times shorter shoulder phase” by “faster resulting in a higher α of 0.39 a lower β of 2.68 producing a two times shorter shoulder phase than at 10 °C”;

Discussion

Line 462, replace “not further analyzed” by “not evaluated”;

Line 480-482, replace “Therefore, our study is the first to analyze the die-off of R. solanacearum phylotype II under anoxic conditions in natural water, and further investigation should elucidate the metabolic pathway to these conditions.” by ““Therefore, our study is the first to analyze the die-off of R. solanacearum phylotype II under anoxic conditions in natural water. Further investigation should elucidate the metabolic activity of R. solanacearum under these conditions.”;

Line 496, replace “at 25 and 10 °C” by “at 25 than at 10 °C”;

Line 496, replace “For the SRP, we also observed a faster decline within 4 days at a higher” by “Whereas for the SRP, we observed a faster decline within 4 days at a higher”;

Line 499-500, replace “The long-term persistence in sterile aquatic microcosms, unaffected by the present microbiota, has been shown earlier for the SRP” by “The long-term persistence in sterile aquatic microcosms, where the indigenous microbiota was absent, has been shown earlier for the SRP”;

Line 505, replace “that it is better adapted” by “that the former is better adapted”;

Line 510, replace “against the microbiota” by “against the existing microbiota”;

Line 511-512, replace “microbiota was only characterized by plating on TSA and R2A medium at 25 °C” by “microbiota was only represented by the culturable community (growing in TSA and R2A, respectively).”;

Line 513, replace “the microbiome” by “microbial community”;

Line 518-519, replace “Interestingly, we observed similar die-off periods in anoxic AW for all the tested pathogens,” by “Interestingly, we observed similar and longer die-off periods in anoxic Aw than in TDW for all the tested pathogens.”;

Line 519-521, replace “even though the culturable concentrations of present background bacteria were comparable in the TDW and AW. Nevertheless, the composition of the microbiota might differ in the AW and TDW, as reported before [52].” by “Although TDW and AW presented comparable concentrations of culturable bacteria the composition of the microbiota might differ between both, as previously reported [52].”;

Line 521-523, replace “There, the authors state that the total diversity of the microbiota in shallow aquifers is lower than in the overlying surface, which supports the different die-off periods of the pathogens in natural TDW and AW, influenced by the present microbiota. The TDW used in our…” by “These authors stated that the total diversity of the microbiota in shallow aquifers is lower than in the overlying surface, which could support the different die-off periods of the pathogens in natural TDW and AW, observed in the present study. In fact, the TDW used in our…”

Line 527, replace “The effects of the different fractions of the native microbiota on the pathogens have not been further explored in this study, but were subjected for R. solanacearum by Alvarez et. al. [22]. There, the native microorganisms altogether (protozoa, bacteria,…” by “Although, the effects of the different fractions of the native microbiota on the pathogens were not accessed in this study, previous studies reported that the native microorganisms (e.g., protozoa, bacteria,…”

Line 534, replace “persistence of the pathogens” by “persistence of the pathogens, under this condition”;

Line 536, replace “products” by “changes”;

Line 537, replace “as well” by “and to improve”;

Line 537-540, replace “To conclude, the introduced pathogens competed better against the AW microbiota than of the TDW, where fewer predators (e.g. protozoa) are present, or where the absence of oxygen can change the metabolic responses of the bacteria which increase their fitness in the AW.” by “The obtained data suggests that the inoculated pathogens competed better against the AW microbiota than to the TDW, where fewer predators (e.g. protozoa) are present, or where the absence of oxygen induces metabolic responses, increasing the pathogen fitness in the AW.”;

Line 542-543, replace “, but this is topic for future research.” by “and needs to be further addressed.”

Line 584, replace “>” by “<”;

Conclusions

Line 618, remove “from”;

Line 619, replace “took between” by “occurred between”;

Line 624-625, replace “increases which will especially affect agricultural production that accounts for about 70% of fresh water use.” by “increases, which will affect especially agricultural production accounting for about 70% of fresh water use”;

Line 625-627, replace “MAR is an option to store fresh TDW in the subsurface for later use as irrigation water but water quality aspects need to be considered as (plant) pathogens may enter the system.” by “MAR is an option to store fresh TDW in the subsurface for later use as irrigation water but water quality, particularly, related to the presence of plant pathogens needs to be considered.”

We look forward to receiving your revised manuscript.

Kind regards,

Ana R. Lopes, PhD

Academic Editor

PLOS ONE

Reviewers' comments:

Reviewer's Responses to Questions

**Comments to the Author**

1. If the authors have adequately addressed your comments raised in a previous round of review and you feel that this manuscript is now acceptable for publication, you may indicate that here to bypass the “Comments to the Author” section, enter your conflict of interest statement in the “Confidential to Editor” section, and submit your "Accept" recommendation.

Reviewer #2: All comments have been addressed

2. Is the manuscript technically sound, and do the data support the conclusions?

Reviewer #2: Yes

3. Has the statistical analysis been performed appropriately and rigorously? 

Reviewer #2: Yes

4. Have the authors made all data underlying the findings in their manuscript fully available?

Reviewer #2: Yes

5. Is the manuscript presented in an intelligible fashion and written in standard English?

Reviewer #2: Yes

6. Review Comments to the Author

Reviewer #2: The updated version has been largely improved, and the authors have addressed my previous concerns. Good work.

7. PLOS authors have the option to publish the peer review history of their article (what does this mean?). If published, this will include your full peer review and any attached files.

Reviewer #2: No

---

## [Author Response · Author response to Decision Letter 1]

29 Mar 2021

Response to Reviewers

PONE-D-20-35253R1

Die-off of plant pathogenic bacteria in tile drainage and anoxic water from a managed aquifer recharge site

PLOS ONE

Dear Dr. Eisfeld,

Thank you for submitting your manuscript to PLOS ONE. After careful consideration, we feel that it has merit but does not fully meet PLOS ONE’s publication criteria as it currently stands. Therefore, we invite you to submit a revised version of the manuscript that addresses the points raised during the review process.

Although the authors have addressed all the previous suggestions, there are still some points that need to be addressed. 

Reply by the authors: 

We thank the editor for the second revision of our manuscript and giving useful comments to improve our paper for publication in PLOS ONE. We are pleased that our first revision answered the questions of the Editor and the Reviewers satisfactorily. Hereby, we would like to address the remaining questions. Replies by the authors are indicated in pink and small textural changes are edited in the manuscript and are declared with “Done” in this rebuttal letter. All changes in the revised manuscript were made using the track changes function in MS Word. It was uploaded together with this rebuttal letter. 

Regarding the culturable indigenous bacteria present in both oxic and anoxic microcosms, the inclusion of table S1 was important. But the duration of the experiments is not clear, based on the days presented in Fig 3 and on table S1. Enumerations in the controls are presented for day 0 and 16, but not all the experiments ended at day 16, right? 

Response by the authors: The enumerations were done at day 0, 12 and 16 post inoculation (dpi) and indeed, not all experiments were concluded at the time point of 16 dpi. The duration of the experiments can be followed in the die-off curves of Figure 3. An exact duration of the sampling of each experiment can be found in the Supplementary File 2 and 3. For clarification, we added the following sentence to Table S1:

“*Controls are non-inoculated water microcosm and the time indication refers to the start of the experiment when pathogen microcosms were inoculated. Data on the total duration of the die off experiments are found in Figure 3.”

Besides, for the inoculated microcosms there is no enumeration of the culturable bacteria at day 0, and also the enumeration at the end of the experiment is missing. The experiments were longer than 12 or 16 days, right?

Response by the authors: The anoxic and oxic control microcosms were prepared with the same water type on the same day as the inoculated ones. Therefore, the inoculated microcosms were not tested at day 0 because they will have the same concentration in culturable bacteria as the controls. After inoculation at day 0, the concentration of culturable bacteria was increased by 104 CFU/mL which refers to the concentration of introduced pathogen. The initial pathogen concentration was only evaluated on day 0 with the semi-selective media supplemented with antibiotics in which the growth of indigenous bacteria was suppressed. No enumeration of culturable bacteria was done at the end of the experiment. Nevertheless, the scope of this study was the die off kinetics and not the specific influence of indigenous microorganism on the pathogens. Enumeration of the culturable bacteria allows to determine the potential effect of the indigenous microbiome on the pathogens, although we realize that this will also depend on the composition of the microbiome. 

Besides, in the material and methods the authors referred line 213-214 “Experiments with natural aquifer water where nitrate was added, were only tested with R. solanacearum and D. solani.”, why not evaluating the die-off of P. carotovorum sp. carotovorum under this condition?

Response by the authors: D. solani and P. carotovorum sp. carotovorum both belong to the family of the soft rot Pectobacteriaceae (SRP). After observing that their die-off of in oxic natural or filtered tile drainage water (TDW) was very similar, we decided to only test one of the SRP for the effect of nitrate. In addition, only a small effect one die off of the D. solani or R. solanacearum was found and therefore we decided to not further extend investigations on this topic. 

Line 165-170, why is this information important?

Response by the authors: During managed aquifer recharge, oxic TDW is infiltrated into an anoxic aquifer which may change the hydrochemistry of the aquifer over a long-term as well as the microbial community of the aquifer. With the information in line 165-170 we want to inform the reader that the anoxic water sample was derived from a relatively (1 year) undisturbed aquifer where a stable hydrochemistry and aquifer community can be found. 

Small issues that need to be addressed: 

Abstract

Line 31 Delete “0.22 μm”; Done.

Line 31 replace “anoxic aquifer water” by “anoxic aquifer water (AW)”; Done.

Line 32 Replace “are” by “were”; Done.

Line 40-41 remove “Higher temperature and the presence of background microbiota strongly

41 accelerated die-off.”; Done.

Line 41, replace “In anoxic natural aquifer” by “Whereas In anoxic natural aquifer”; Done.

Line 44-46, replace “The non-linear model provides a tool to reliably predict the die-off of plant pathogenic bacteria or other pathogenic microorganisms in the context of microbial risk assessment.”

by “The non-linear model was shown to be a reliable tool to predict the die-off of the analyzed plant pathogenic bacteria, suggesting its further application to other pathogenic microorganisms in the context of microbial risk assessment.”; Done.

Material and Methods

Table 1, move the row “treatment” to the top of the table, above the row water type; Done.

Line 209-211, replace “For both the natural microcosms of oxic TDW and anoxic AW, a respective control microcosm that was not inoculated with pathogens was prepared” by “For both the natural microcosms of oxic TDW and anoxic AW a non-inoculated control microcosm was prepared. Done.

Line 214-218, replace “The temperatures were chosen as they represent, at the lower end, the nearly constant temperature in the aquifer (10 °C); while the upper end (25 °C) reflects infiltration of rainfall events during the warmer summer months. Moreover, it covers a temperature that is more representative for tropical regions, where these bacteria also play an important role as disease causing organisms.” 

By “The temperatures were chosen as they represent the nearly constant temperature in the aquifer (10 °C); while 25 °C reflects infiltration of rainfall events during the warmer summer months. Moreover, the last covers a temperature more representative of tropical regions, where these bacteria also play an important role as disease causing organisms.”; Done.

Line 220, replace “microbiome” by “microbial”; Done.

Line 221, remove “a sterile filter holder with receiver (Nalgene, USA) and”; Done.

Line 265, replace “unpublished results” by “data not shown”; Done.

Line 266/511/, replace “background” by “indigenous”; Done.

Line 279, replace by “the bacterial” by “the pathogen”; Done.

Figure 2, replace by “the population concentration” by “the pathogen concentration”; Done.

Results

Line 363/367, replace “S1 Table” by “Table S2”; Done.

Line 370, replace “in S1 Fig” by “Fig S1”; Done.

Line 443, replace “faster resulting in a higher α of 0.39, and β is 2.68 producing a two times shorter shoulder phase” by “faster resulting in a higher α of 0.39 a lower β of 2.68 producing a two times shorter shoulder phase than at 10 °C”; Done.

Discussion

Line 462, replace “not further analyzed” by “not evaluated”; Done.

Line 480-482, replace “Therefore, our study is the first to analyze the die-off of R. solanacearum phylotype II under anoxic conditions in natural water, and further investigation should elucidate the metabolic pathway to these conditions.” by ““Therefore, our study is the first to analyze the die-off of R. solanacearum phylotype II under anoxic conditions in natural water. Further investigation should elucidate the metabolic activity of R. solanacearum under these conditions.”; Done.

Line 496, replace “at 25 and 10 °C” by “at 25 than at 10 °C”; Done.

Line 496, replace “For the SRP, we also observed a faster decline within 4 days at a higher” by “Whereas for the SRP, we observed a faster decline within 4 days at a higher”; Done.

Line 499-500, replace “The long-term persistence in sterile aquatic microcosms, unaffected by the present microbiota, has been shown earlier for the SRP” by “The long-term persistence in sterile aquatic microcosms, where the indigenous microbiota was absent, has been shown earlier for the SRP”; Done.

Line 505, replace “that it is better adapted” by “that the former is better adapted”; Done.

Line 510, replace “against the microbiota” by “against the existing microbiota”; Done.

Line 511-512, replace “microbiota was only characterized by plating on TSA and R2A medium at 25 °C” by “microbiota was only represented by the culturable community (growing in TSA and R2A, respectively).”; Done.

Line 513, replace “the microbiome” by “microbial community”; Done.

Line 518-519, replace “Interestingly, we observed similar die-off periods in anoxic AW for all the tested pathogens,” by “Interestingly, we observed similar and longer die-off periods in anoxic Aw than in TDW for all the tested pathogens.”; Done.

Line 519-521, replace “even though the culturable concentrations of present background bacteria were comparable in the TDW and AW. Nevertheless, the composition of the microbiota might differ in the AW and TDW, as reported before [52].” by “Although TDW and AW presented comparable concentrations of culturable bacteria the composition of the microbiota might differ between both, as previously reported [52].”; Done.

Line 521-523, replace “There, the authors state that the total diversity of the microbiota in shallow aquifers is lower than in the overlying surface, which supports the different die-off periods of the pathogens in natural TDW and AW, influenced by the present microbiota. The TDW used in our…” by “These authors stated that the total diversity of the microbiota in shallow aquifers is lower than in the overlying surface, which could support the different die-off periods of the pathogens in natural TDW and AW, observed in the present study. In fact, the TDW used in our…” Done.

Line 527, replace “The effects of the different fractions of the native microbiota on the pathogens have not been further explored in this study, but were subjected for R. solanacearum by Alvarez et. al. [22]. There, the native microorganisms altogether (protozoa, bacteria,…” by “Although, the effects of the different fractions of the native microbiota on the pathogens were not accessed in this study, previous studies reported that the native microorganisms (e.g., protozoa, bacteria,…” Done.

Line 534, replace “persistence of the pathogens” by “persistence of the pathogens, under this condition”; Done.

Line 536, replace “products” by “changes”; Done.

Line 537, replace “as well” by “and to improve”; Done.

Line 537-540, replace “To conclude, the introduced pathogens competed better against the AW microbiota than of the TDW, where fewer predators (e.g. protozoa) are present, or where the absence of oxygen can change the metabolic responses of the bacteria which increase their fitness in the AW.” by “The obtained data suggests that the inoculated pathogens competed better against the AW microbiota than to the TDW, where fewer predators (e.g. protozoa) are present, or where the absence of oxygen induces metabolic responses, increasing the pathogen fitness in the AW.”; Done.

Line 542-543, replace “, but this is topic for future research.” by “and needs to be further addressed.” Done.

Line 584, replace “>” by “<”; Done.

Conclusions

Line 618, remove “from”; Done.

Line 619, replace “took between” by “occurred between”; Done.

Line 624-625, replace “increases which will especially affect agricultural production that accounts for about 70% of fresh water use.” by “increases, which will affect especially agricultural production accounting for about 70% of fresh water use”; Done.

Line 625-627, replace “MAR is an option to store fresh TDW in the subsurface for later use as irrigation water but water quality aspects need to be considered as (plant) pathogens may enter the system.” by “MAR is an option to store fresh TDW in the subsurface for later use as irrigation water but water quality, particularly, related to the presence of plant pathogens needs to be considered.” Done.

---

## [Editor Report · Decision Letter 2]

6 Apr 2021

Die-off of plant pathogenic bacteria in tile drainage and anoxic water from a managed aquifer recharge site

PONE-D-20-35253R2

Dear Dr. Eisfeld,

We’re pleased to inform you that your manuscript has been judged scientifically suitable for publication and will be formally accepted for publication once it meets all outstanding technical requirements.

Besides, I would like to add a few changes. Verify if the last submission of table S1 is the more recent as the sentence referred in the response is not in the submitted file. “For clarification, we added the following sentence to Table S1:“*Controls are non-inoculated water microcosm and the time indication refers to the start of the experiment when pathogen microcosms were inoculated. Data on the total duration of the die off experiments are found in Figure 3.””.

Remove sentence line 129-130 “It takes” and replace the word “background” by “indigenous” line 330, 335. Additionally, within one week, you’ll receive an e-mail detailing further required amendments. When these have been addressed, you’ll receive a formal acceptance letter and your manuscript will be scheduled for publication.

Kind regards,

Ana R. Lopes, PhD

Academic Editor

PLOS ONE
---

## [Editor Report · Acceptance letter]

26 Apr 2021

PONE-D-20-35253R2 

Die-off of plant pathogenic bacteria in tile drainage and anoxic water from a managed aquifer recharge site 

Dear Dr. Eisfeld:

I'm pleased to inform you that your manuscript has been deemed suitable for publication in PLOS ONE. Congratulations! Your manuscript is now with our production department. 

Kind regards, 

on behalf of

Dr. Ana R. Lopes 

Academic Editor

PLOS ONE